# Hybrid immunity improves B cells and antibodies against SARS-CoV-2 variants

Emanuele Andreano[1], Ida Paciello[1], Giulia Piccini[2], Noemi Manganaro[1], Piero Pileri[1], Inesa Hyseni[2,3], Margherita Leonardi[2,3], Elisa Pantano[1], Valentina Abbiento[1], Linda Benincasa[3], Ginevra Giglioli[3], Concetta De Santi[1], Massimiliano Fabbiani[4], Ilaria Rancan[4,5], Mario Tumbarello[4,5], Francesca Montagnani[4,5], Claudia Sala[1], Emanuele Montomoli[2,3,6] & Rino Rappuoli[1,7 ✉]

The emergence of SARS-CoV-2 variants is jeopardizing the effectiveness of current vaccines and limiting the application of monoclonal antibody-based therapy for COVID-19 (refs. [1,2]). Here we analysed the memory B cells of five naive and five convalescent people vaccinated with the BNT162b2 mRNA vaccine to investigate the nature of the B cell and antibody response at the single-cell level. Almost 6,000 cells were sorted, over 3,000 cells produced monoclonal antibodies against the spike protein and more than 400 cells neutralized the original SARS-CoV-2 virus first identified in Wuhan, China. The B.1.351 (Beta) and B.1.1.248 (Gamma) variants escaped almost 70% of these antibodies, while a much smaller portion was impacted by the B.1.1.7 (Alpha) and B.1.617.2 (Delta) variants. The overall loss of neutralization was always significantly higher in the antibodies from naive people. In part, this was due to the IGHV2-5;IGHJ4-1 germline, which was found only in people who were convalescent and generated potent and broadly neutralizing antibodies. Our data suggest that people who are seropositive following infection or primary vaccination will produce antibodies with increased potency and breadth and will be able to better control emerging SARS-CoV-2 variants.

Twenty months after the beginning of the COVID-19 pandemic, with 252 million people infected, 5 million deaths and 7.2 billion vaccine doses administered, the world is still struggling to control the virus. In most developed countries, vaccines have vastly reduced severe disease, hospitalization and deaths, but they have not been able to control the infections that are fuelled by new and more infectious variants. A large number of studies so far have shown that protection from infection is linked to the production of neutralizing antibodies against the spike (S) protein of the virus[3–6]. This is a metastable, trimeric class 1 fusion glycoprotein, composed of the S1 and S2 subunits, and mediates virus entry, changing from a prefusion to postfusion conformation after binding to the human angiotensin-converting enzyme 2 (ACE2) receptor and heparan sulfates on the host cells[7]. Potent neutralizing antibodies recognize the S1 subunit of each monomer, which includes the receptor-binding domain (RBD) and N-terminal domain (NTD) immunodominant sites[8]. The large majority of neutralizing antibodies bind to the receptor-binding motif, within the RBD, and a smaller fraction targets the NTD[5,9]. Neutralizing antibodies against the S2 subunit have been described; however, they have very low potency[5,10]. Neutralizing antibodies generated after infection derive in large part from germline IGHV3-53 and the closely related IGHV3-66 with very few somatic mutations[11,12]. From June 2020, the virus started to generate mutations that allowed the virus to evade neutralizing antibodies, to become more

infectious, or both. Some of the mutant viruses completely replaced the original SARS-CoV-2 first detected in Wuhan, China. The most successful variant viruses are B.1.1.7 (Alpha), B.1.351 (Beta), B.1.1.248 (Gamma) and B.1.617.2 (Delta), which have been named variants of concern (VoCs)[13]. The Delta variant is currently spreading across the globe and causing large concerns also in fully vaccinated populations. It is therefore imperative to understand the molecular mechanisms of the immune response to vaccination to design better vaccines and vaccination policies. Several investigators have shown that vaccination of people who are convalescent can yield neutralizing antibodies that can be up to a thousand-fold higher than those induced by infection or vaccination, suggesting that one way of controlling the pandemic may be the induction of a hybrid immunity-like response using a third booster dose[14–18]. At the single-cell level, here we compared the nature of the neutralizing antibody response against the original virus first detected in Wuhan and the VoCs in naive and convalescent participants who were immunized with the BNT162b2 mRNA vaccine.

## B cell response in COVID-19 vaccinees

We enrolled ten donors who were vaccinated with the BNT162b2 mRNA vaccine: five of them were healthy people who were naive to SARS-CoV-2 infection at vaccination (seronegative) and the other five had recovered

[1]Monoclonal Antibody Discovery (MAD) Lab, Fondazione Toscana Life Sciences, Siena, Italy. [2]VisMederi S.r.l, Siena, Italy. [3]VisMederi Research S.r.l., Siena, Italy. [4]Department of Medical Sciences, Infectious and Tropical Diseases Unit, Siena University Hospital, Siena, Italy. [5]Department of Medical Biotechnologies, University of Siena, Siena, Italy. [6]Department of Molecular and Developmental Medicine, University of Siena, Siena, Italy. [7]Department of Biotechnology, Chemistry and Pharmacy, University of Siena, Siena, Italy. ✉e-mail: rino.r.rappuoli@gsk.com

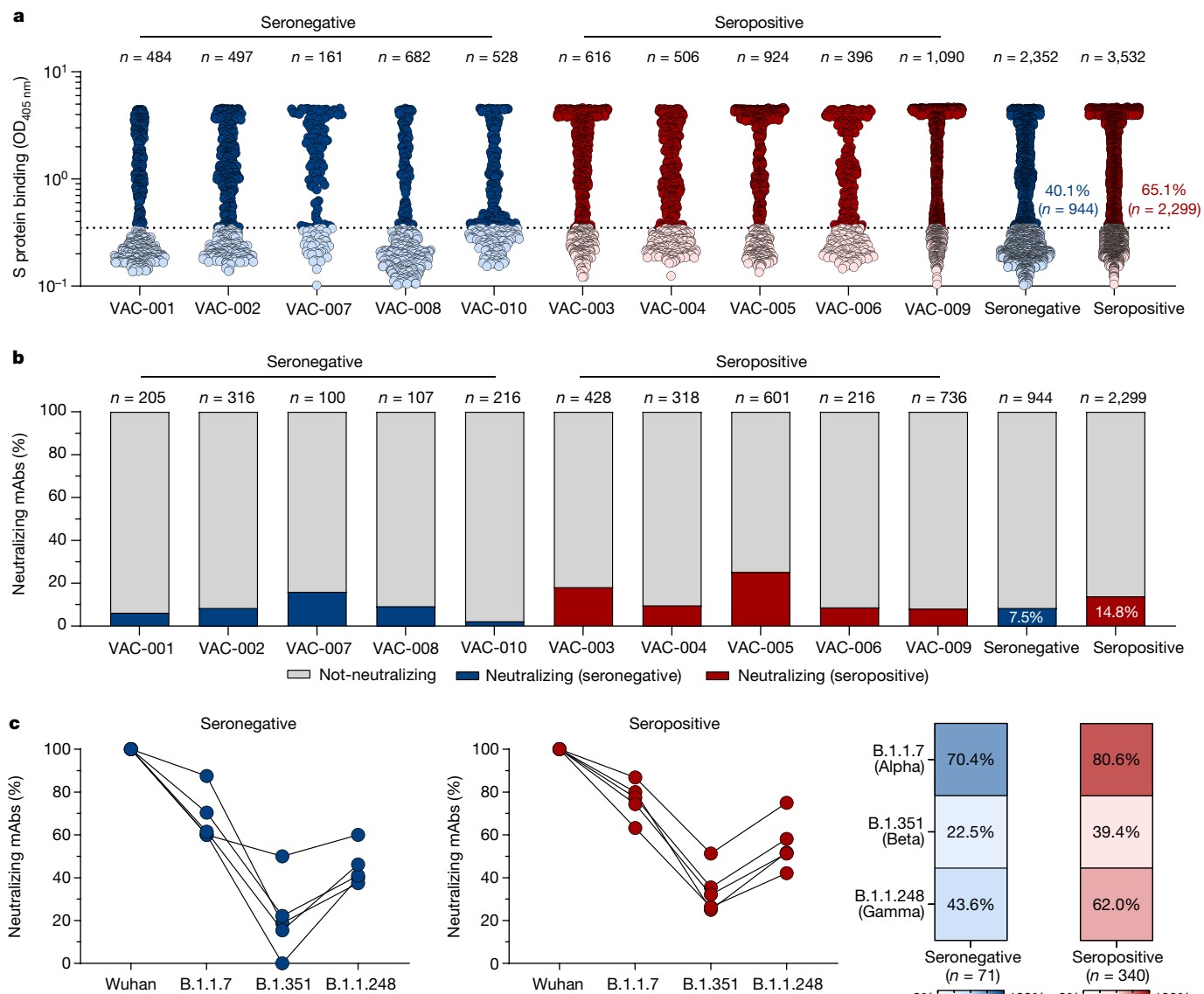

**Fig. 1 | Identification of cross-neutralizing SARS-CoV-2 S protein-specific nAbs. a**, The graph shows supernatants that were tested for binding to the SARS-CoV-2 S protein antigen first detected in Wuhan, China. The threshold of positivity was set as two times the value of the blank (dotted line). The dark blue and dark red dots represent mAbs that bind to the S protein for vaccinees who were seronegative and seropositive, respectively. The light blue and light red dots represent mAbs that do not bind to the S protein for vaccinees who were seronegative and seropositive, respectively. OD, optical density. **b**, The bar graph shows the percentage of not-neutralizing antibodies (grey), nAbs from

individuals who were seronegative (dark blue) and nAbs for individuals who were seropositive (dark red). The total number (*n*) of antibodies tested per individual is shown on the top of each bar in **a**, **b**. **c**, The graphs show the fold-change percentage of nAbs in individuals who were seronegative (left) and seropositive (right) against the Alpha, Beta and Gamma VoCs compared with the original SARS-CoV-2 virus detected in Wuhan. The heat maps show the overall percentage of the SARS-CoV-2 nAbs detected in Wuhan that are able to neutralize the tested VoCs.

from SARS-CoV-2 infection before vaccination (seropositive). Participant details are summarized in Extended Data Table 1. Blood collection occurred at an average of 48 and 21 days after the last vaccination dose for participants who were seronegative and seropositive, respectively (Extended Data Table 1). This difference may affect the frequency of circulating B cells and the serum activity of participants who are seronegative and seropositive analysed in this study. We initially analysed the frequency of circulating B cell populations between our groups. Participants who were seropositive showed a 2.46-fold increase in S-protein-specific CD19+CD27+IgD−IgM− memory B cells compared with participants who were seronegative and an overall 10% higher count of CD19+CD27+IgD−IgM− memory B cells (Extended Data Fig. 1a–c). Conversely, participants who were seronegative showed a 2.3-fold higher frequency of CD19+CD27+IgD−IgM+ memory B cells than participants

who were seropositive. No differences were found in the numbers of CD19+CD27+IgD−IgM+ S protein+ memory B cells between the two groups assessed in this study (Extended Data Fig. 1a–c). Following the analyses of memory B cells, we characterized the polyclonal response of these donors by testing their binding response to the S protein trimer, RBD, NTD and the S2 domain, and subsequently by testing their neutralization activity against the original SARS-CoV-2 virus first detected in Wuhan (Extended Data Fig. 2). Plasma from participants who were seropositive showed a higher binding activity to the S protein and all tested domains than plasma from participants who were seronegative (Extended Data Fig. 2a–d). In addition, participants who were seropositive showed a tenfold-higher neutralization activity against the original SARS-CoV-2 virus detected in Wuhan than in participants who were seronegative (Extended Data Fig. 2e, f).

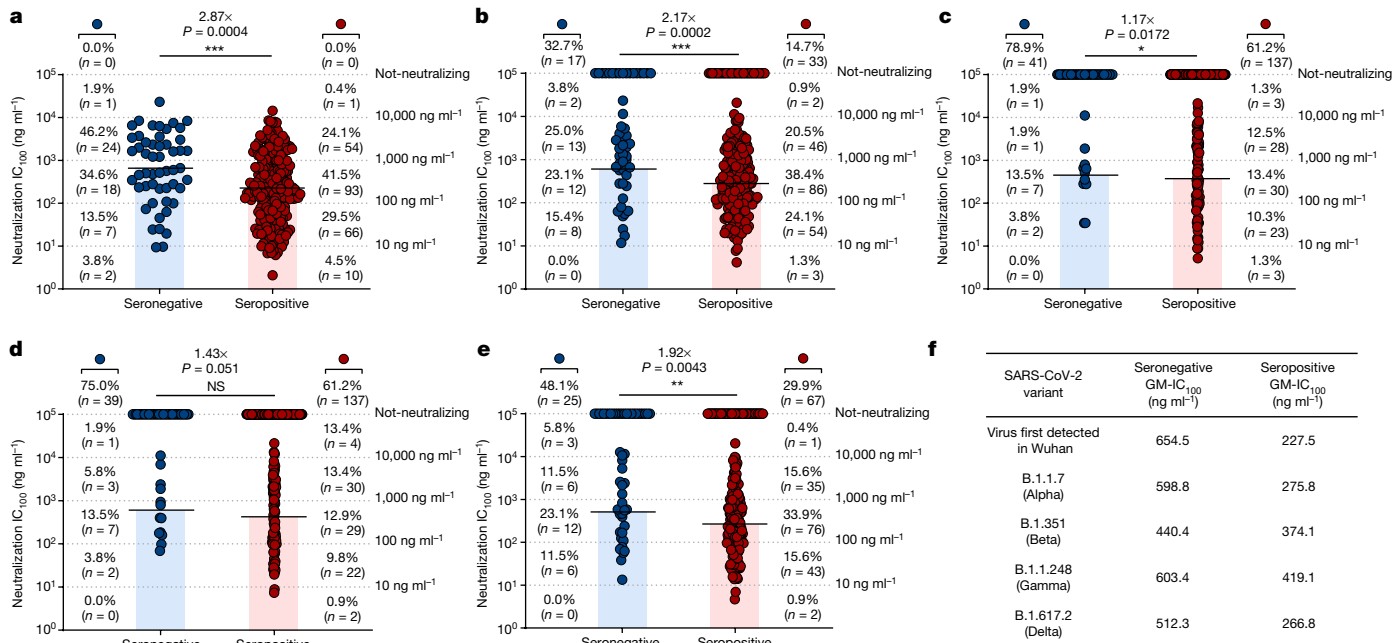

**Fig. 2 | Potency and breadth of neutralization of nAbs against SARS-CoV-2 and VoCs. a–e**, Scatter dot charts show the neutralization potency, reported as $IC_{100}$ (ng ml$^{-1}$), of nAbs tested against the original SARS-CoV-2 virus first detected in Wuhan (**a**) and the B.1.1.7 (**b**), B.1.351 (**c**), B.1.1.248 (**d**) and B.1.617.2 (**e**) VoCs. The number and percentage of nAbs from individuals who were seronegative versus seropositive, fold change, neutralization $IC_{100}$ geometric mean (black lines, blue and red bars) and statistical significance are denoted on each graph. A non-parametric Mann–Whitney $t$-test was used to evaluate statistical significances between groups. Two-tailed $P$ value significances are shown as *$P < 0.05$, **$P < 0.01$, ***$P < 0.001$. NS, not significant. **f**, The table shows the $IC_{100}$ geometric mean (GM) of all nAbs pulled together from each group against all SARS-CoV-2 viruses tested. Technical duplicates were performed for each experiment.

## Isolation of neutralizing antibodies

To better characterize the B cell immune response, we single-cell-sorted antigen-specific memory B cells using the SARS-CoV-2 S protein antigen identified in Wuhan as bait, which was encoded by the mRNA vaccine. The single-cell sorting strategy was performed as previously described[5]. In brief, the prefusion S protein trimer-specific (S protein$^+$), class-switched memory B cells (CD19$^+$CD27$^+$IgD$^-$IgM$^-$) were single-cell-sorted and then incubated for 2 weeks to naturally produce and release monoclonal antibodies (mAbs) into the supernatant. A total of 2,352 and 3,532 S protein$^+$ memory B cells were sorted from vaccinees who were seronegative and seropositive, respectively (Extended Data Table 2). Of these, 944 (40.1%) and 2,299 (65.1%), respectively, were released in the supernatant mAbs, recognizing the S protein prefusion trimer in enzyme-linked immunosorbent assay (ELISA) (Fig. 1a, Extended Data Table 2). These mAbs were then tested in a cytopathic effect-based microneutralization assay (CPE-MN) with the original live SARS-CoV-2 virus detected in Wuhan at a single point dilution (1:5) to identify SARS-CoV-2 neutralizing human monoclonal antibodies (nAbs). This first screening identified a total of 411 nAbs, of which 71 derived from participants who were seronegative and 340 were from participants who were seropositive (Fig. 1b, Extended Data Table 2). Overall, the fraction of S-protein-specific B cells producing nAbs were 7.5% for participants who were seronegative and 14.8% for participants who were seropositive. Following this first screening, all nAbs that were able to neutralize the SARS-CoV-2 virus detected in Wuhan were tested by CPE-MN against major VoCs, including B.1.1.7 (Alpha), B.1.351 (Beta) and B.1.1.248 (Gamma) to understand the breadth of neutralization of nAbs elicited by the BNT162b2 mRNA vaccine. At the time of this assessment the B.1.617.2 (Delta) variant had not yet spread globally and therefore was not available for screening. Participants who were seropositive had an overall higher percentage of nAbs neutralizing the VoCs than participants who were seronegative. The average frequency

of nAbs from participants who were seropositive neutralizing the Alpha, Beta and Gamma variants was 80.6% ($n = 274$), 39.4% ($n = 134$) and 62.0% ($n = 211$), respectively, compared with 70.4% ($n = 50$), 22.5% ($n = 16$) and 43.6% ($n = 31$), respectively, in participants who were seronegative (Fig. 1c, Extended Data Table 2).

## Potency and breadth against variants

To better characterize and understand the potency and breadth of coverage of all nAbs against the SARS-CoV-2 virus detected in Wuhan, we aimed to express all the 411 nAbs previously identified as IgG1. We were able to recover and express 276 antibodies for further characterization, 224 (89.8%) from participants who were seropositive and 52 (10.2%) from participants who were seronegative. Initially, antibodies were tested for binding against the RBD, NTD and the S2 domain of the original SARS-CoV-2 S protein identified in Wuhan. Overall, no major differences were observed in nAbs that recognized the RBD and NTD, whereas nAbs that were able to bind to the S protein only in its trimeric conformation (that is, not able to bind single domains) were almost threefold higher in participants who were seronegative than in participants who were seropositive (Extended Data Fig. 3). None of the tested nAbs targeted the S2 domain. nAbs were then tested by CPE-MN in serial dilution to evaluate their 100% inhibitory concentration ($IC_{100}$) against the SARS-CoV-2 virus detected in Wuhan and the VoCs. At this stage of the study, the B.1.617.2 (Delta) virus had spread globally, and we were able to obtain the live virus for our experiments. Overall, nAbs isolated from vaccinees who were seropositive had a significantly higher potency than those isolated from vaccinees who were seronegative. The $IC_{100}$ geometric mean in participants who were seropositive was 2.87-fold, 2.17-fold, 1.17-fold, 1.43-fold and 1.92-fold lower than in participants who were seronegative for the virus detected in Wuhan, and the Alpha, Beta, Gamma and Delta VoCs, respectively (Fig. 2). In addition, a bigger fraction of nAbs from participants who were seropositive retained the

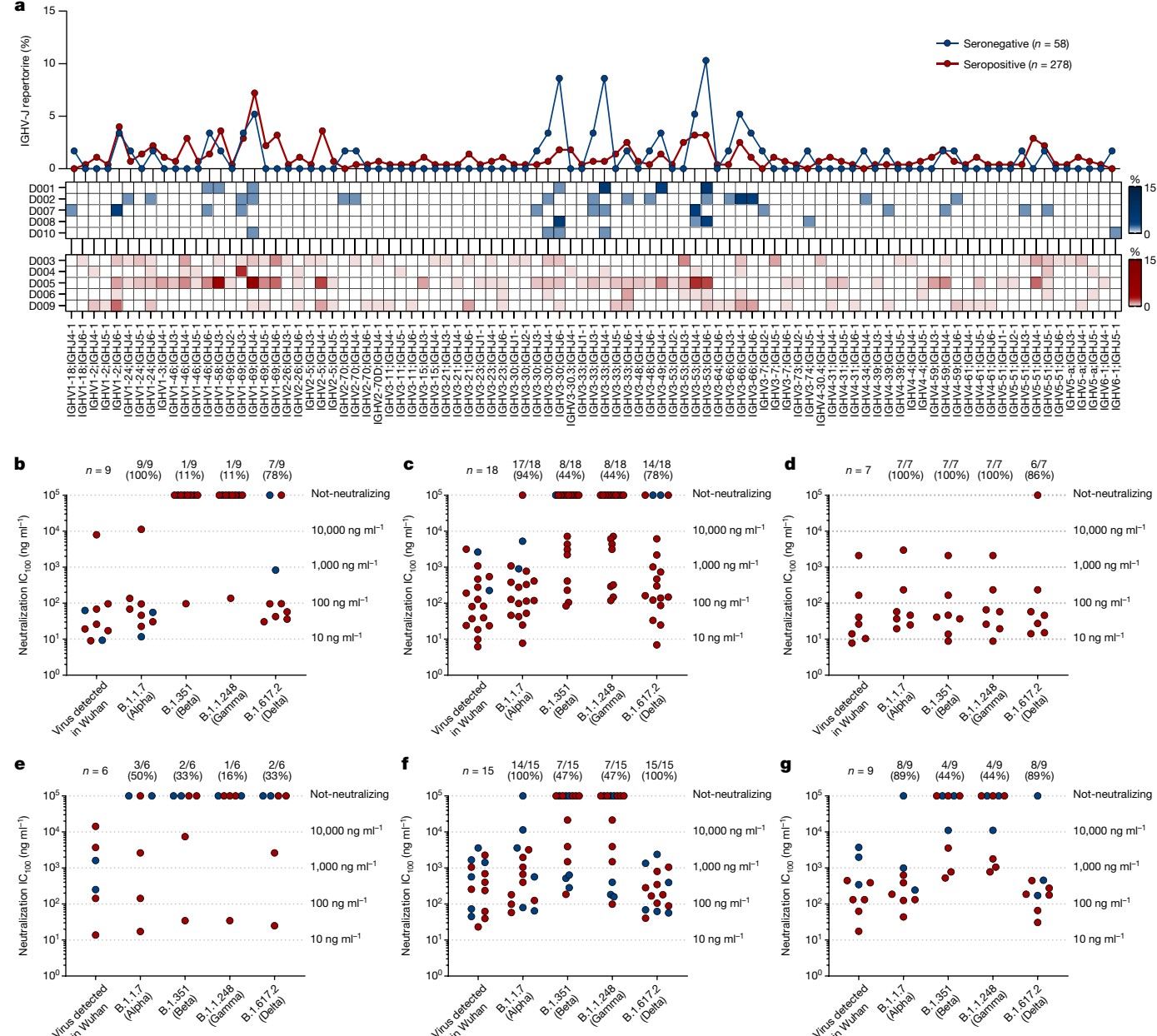

**Fig. 3 | Repertoire analyses and functional characterization of predominant gene-derived nAbs. a**, The graph shows the IGHV-J rearrangement frequencies between vaccinees who were seronegative and seropositive (top), and the frequency within seronegative (middle) and seropositive (bottom) participants. **b–g**, The graphs show the neutralization potency ($IC_{100}$) of predominant gene-derived nAbs from the IGHV1-2;IGHJ6-1 (**b**), IGHV1-69;IGHJ4-1 (**c**), IGHV2-5;IGHJ4-1 (**d**), IGHV3-30;IGHJ6-1 (**e**), IGHV3-53;IGHJ6-1 (**f**) and IGHV3-66;IGHJ4-1 (**g**) families, against the original SARS-CoV-2 virus first detected in Wuhan and the B.1.1.7, B.1.351, B.1.1.248 and B.1.617.2 VoCs.

ability to neutralize the VoCs. Indeed, when nAbs were individually tested against all VoCs, the ability to neutralize the Alpha, Beta, Gamma and Delta variants was lost by 14%, 61%, 61% and 29% of the antibodies from participants who were seropositive versus 32%, 78%, 75% and 46% of those from participants who were seronegative, respectively (Fig. 2). Finally, a major difference between participants who were seronegative and seropositive was found in the class of nAbs with medium/high potency ($IC_{100}$ of 11–100 ng ml$^{-1}$ and 101–1,000 ng ml$^{-1}$) against all variants. Indeed, nAbs in these ranges from participants who were seropositive constitute 71.0%, 62.5%, 23.7%, 22.8% and 53.1% of the whole nAb repertoire, whereas nAbs from seronegative donors were 48.1%, 38.5%, 17.3%, 17.3% and 34.6% against the SARS-CoV-2 virus detected in Wuhan and the Alpha, Beta, Gamma and Delta VoCs respectively (Fig. 2).

## Functional gene repertoire

The analysis of the immunoglobulin G heavy chain variable (IGHV) and joining (IGHJ) gene rearrangements of 58 and 278 sequences recovered from participants who were seronegative and seropositive, respectively, showed that they use a broad range of germlines and share the most abundant germlines. In particular, both groups predominantly used the IGHV1-69;IGHJ4-1 and IGHV3-53;IGHJ6-1 germlines, which were shared by three out five participants per group (Fig. 3a). In addition, the IGHV3-30;IGHJ6-1 and IGHV3-33;IGHJ4-1 germlines, which were more abundant in donors who were seronegative, and the IGHV1-2;IGHJ6-1 germline, which was mainly expanded in vacinees who were seropositive, were also used with high frequency in both groups. Only the IGHV2-5;IGHJ4-1

germline was seen to be predominantly expanded only in donors who were seropositive (Fig. 3a). Despite the fact that selected germlines were boosted following vaccination, no major clonal families were identified, and the biggest family observed contained only four antibodies. To better characterize these predominant gene families, we evaluated their neutralization potency and breadth against SARS-CoV-2 and VoCs. In this analysis, we could not evaluate IGHV3-33;IGHJ4-1 nAbs, as only three of these antibodies were expressed, but we included the IGHV3-53 closely related family IGHV3-66;IGHJ4-1, as this family was previously described to be mainly involved in the neutralization of the SARS-CoV-2 virus[11,19]. A large part of nAbs deriving from these predominant germlines had a very broad range of neutralization potency against the original SARS-CoV-2 virus detected in Wuhan, with the $IC_{100}$ spanning from less than 10 to over 10,000 ng ml$^{-1}$ (Fig. 3b–g). However, many of them lost the ability to neutralize SARS-CoV-2 VoCs. The loss of neutralizing activity occurred for most germlines and it was moderate against the Alpha and Delta variants, whereas the loss was marked against the Beta and Gamma variants (Fig. 3b–g). A notable exception was the IGHV2-5;IGHJ4-1 germline, which was present only in nAbs of participants who were seropositive, that showed potent antibodies able to equally neutralize all SARS-CoV-2 VoCs (Fig. 3d). Finally, we evaluated the CDRH3 length and V-gene somatic hypermutation levels for all nAbs retrieved from participants who were seronegative and seropositive and for predominant germlines. Overall, the two groups show a similar average CDRH3 length (15.0 amino acids and 15.1 amino acids for participants who were seronegative and seropositive, respectively); however, participants who were seropositive showed almost twofold-higher V-gene mutation levels than participants who were seronegative (Extended Data Fig. 4). As for predominant gene-derived nAbs, we observed heterogenous CDRH3 length, with the only exception of IGHV3-53;IGHJ6-1 nAbs, and higher V-gene mutation levels in predominant germlines from participants who were seropositive than in germlines from participants who were seronegative (Extended Data Fig. 5).

## S protein epitope mapping

To map the regions of the S protein recognized by the identified nAbs, we used a competition assay with four known antibodies: J08, which targets the top loop of the receptor-binding motif[5]; S309, which binds to the RBD but outside of the RMB region[20]; 4A8, which recognizes the NTD[21]; and L19, which binds to the S2 domain[5] (Extended Data Fig. 6). The nAbs identified in this study were pre-incubated with the original SARS-CoV-2 S protein detected in Wuhan, and subsequently the four nAbs labelled with different fluorophores were added as a single mix. For one of the four fluorescently labelled nAbs, 50% signal reduction was used as a threshold for positive competition. The vast majority of nAbs from both seronegative (50.0%; $n = 26$) and seropositive (51.3%; $n = 115$) vaccinees competed with J08 (Extended Data Fig. 7a, Extended Data Table 3). For vaccinees who were seronegative, the second most abundant population was composed of nAbs that did not compete with any of the four fluorescently labelled nAbs (25.0%; $n = 13$), followed by nAbs targeting the NTD (17.3%; $n = 9$). As for vaccinees who were seropositive, the second most abundant population was composed of nAbs that competed with S309 (21.4%; $n = 48$), followed by nAbs that competed with 4A8 (15.6%; $n = 35$) and not-competing nAbs (11.6%; $n = 26$). None of our nAbs competed with the S2-targeting antibody L19 (Extended Data Fig. 7a, Extended Data Table 3). nAbs that competed with J08, which are likely to bind to the receptor-binding motif, derived from several germlines, including the predominant IGHV3-53;IGHJ6-1 (10.6%; $n = 14$), IGHV1-69;IGHJ4-1 (8.3%; $n = 11$) and IGHV1-2;IGHJ6-1 (6.8%; $n = 9$) germlines (Extended Data Fig. 7b). By contrast, those that competed with S309 derived mostly from the IGHV2-5;IGHJ4-1 germline (13.7%; $n = 7$), which were isolated exclusively from vaccinees who were seropositive (Extended Data Fig. 7c). As for NTD-directed nAbs, the non-predominant gene family IGHV1-24;IGHJ6-1 was the

most abundant, confirming what was reported in previous studies[22] (Extended Data Fig. 7d). Finally, for nAbs that did not compete with any of the known antibodies used in our competition assay, the non-predominant gene families IGHV1-69;IGHJ3-1 (9.7%; $n = 3$) and IGHV1-69;IGHJ6-1 (9.7%; $n = 3$) were the most abundant (Extended Data Fig. 7e).

## Discussion

Our study analysed the repertoire of B cells producing neutralizing antibodies following vaccination of naive and previously infected people at the single-cell level. The most important conclusion from this work is that people who are previously exposed to SARS-CoV-2 infection respond to vaccination with more B-cell-producing antibodies that are not susceptible to escape variants and that have higher neutralization potency. This can be explained in part by the increased number of somatic mutations and by the fact that participants who are seropositive expand potent antibodies derived from the IGHV2-5;IGHJ4-1 germline, which were not described in naive vaccinees[18]. One limitation of our study is that we did not include people who received a third booster dose of vaccine. Despite this limitation, we believe that our conclusions are likely to be extendable to people who are seronegative, as a third vaccine dose could lead to a hybrid immunity-like response as neutralizing antibodies following infection and vaccination derive mostly from the same immunodominant germlines[11,12,17–19]. Our analysis suggests that a booster dose of vaccine will increase the frequency of memory B cells producing potent neutralizing antibodies that are not susceptible to escape variants and will allow better control of this pandemic. The massive variant escape from predominant germlines, such as IGHV3-53, IGHV3-66, IGHV3-30 and IGHV1-69, and the presence of antibodies deriving from the IGHV2-5 germline that are resistant to variants, suggest that the design of vaccines that preferentially promote or avoid the expansion of selected germlines can generate broad protection against SARS-CoV-2 variants. Germline-targeting vaccination, which has been pioneered in the HIV field[23,24], may be a promising strategy to fight the COVID-19 pandemic.

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

## Methods

### Enrolment of COVID-19 vaccinees and human sample collection

This work results from a collaboration with the Azienda Ospedaliera Universitaria Senese, Siena (IT) that provided samples from donors vaccinated against COVID-19, of both sexes, who gave their written consent. The study was approved by the Comitato Etico di Area Vasta Sud Est (CEAVSE) ethics committees (Parere 17065 in Siena) and conducted according to good clinical practice in accordance with the declaration of Helsinki (European Council 2001, US Code of Federal Regulations, ICH 1997). This study was unblinded and not randomized. No statistical methods were used to predetermine sample size.

### Single-cell sorting of SARS-CoV-2 S protein[+] memory B cells from COVID-19 vaccinees

Peripheral blood mononuclear cells (PBMCs) and the single-cell sorting strategy were performed as previously described[5]. In brief, PBMCs were isolated from heparin-treated whole blood by density gradient centrifugation (Ficoll-Paque PREMIUM, Sigma-Aldrich). After separation, PBMCs were stained with Live/Dead Fixable Aqua (Invitrogen; Thermo Scientific) diluted 1:500 at room temperature. After 20 min of incubation, cells were washed with PBS and unspecific bindings were saturated with 20% normal rabbit serum (Life Technologies). Following 20 min of incubation at 4 °C, cells were washed with PBS and stained with SARS-CoV-2 S protein labelled with Strep-Tactin XT DY-488 (2-1562-050, iba-lifesciences) for 30 min at 4 °C. After incubation, the following staining mix was used CD19 V421 (1:320; 562440, BD), IgM PerCP-Cy5.5 (1:50; 561285, BD), CD27 PE (1:30; 340425, BD), IgD-A700 (1:15; 561302, BD), CD3 PE-Cy7 (1:100; 300420, BioLegend), CD14 PE-Cy7 (1:320; 301814, BioLegend), CD56 PE-Cy7 (1:80; 318318, BioLegend) and cells were incubated at 4 °C for an additional 30 min. Stained memory B cells were single-cell-sorted with a BD FACS Aria III (BD Biosciences) into 384-well plates containing 3T3-CD40L feeder cells and were incubated with IL-2 and IL-21 for 14 days as previously described[25].

### ELISA assay with SARS-CoV-2 S protein prefusion trimer

mAbs and plasma binding specificity against the S protein trimer was detected by ELISA as previously described[5]. In brief, 384-well plates (microplate clear, Greiner Bio-one) were coated with 3 μg/ml of streptavidin (Thermo Fisher) diluted in carbonate-bicarbonate buffer (E107, Bethyl Laboratories) and incubated at room temperature overnight. The next day, plates were incubated for 1 h at room temperature with 3 μg/ml of SARS-CoV-2 S protein diluted in PBS. Plates were then saturated with 50 μl per well of blocking buffer (phosphate-buffered saline and 1% BSA) for 1 h at 37 °C. After blocking, 25 μl per well of mAbs diluted 1:5 in sample buffer (phosphate-buffered saline, 1% BSA and 0.05% Tween-20) was added to the plates and was incubated at 37 °C. Plasma samples derived from vaccinees were tested (starting dilution of 1:10; step dilution of 1:2 in sample buffer) in a final volume of 25 μl per well and were incubated at 37 °C. After 1 h of incubation, 25 μl per well of alkaline phosphatase-conjugated goat anti-human IgG and IgA (Southern Biotech) diluted 1:2,000 in sample buffer was added. Finally, S protein binding was detected using 25 μl per well of PNPP (p-nitrophenyl phosphate; Thermo Fisher) and the reaction was measured at a wavelength of 405 nm by the Varioskan Lux Reader (Thermo Fisher Scientific). After each incubation step, plates were washed three times with 100 μl per well of washing buffer (phosphate-buffered saline and 0.05% Tween-20). Sample buffer was used as a blank and the threshold for sample positivity was set at twofold the optical density (OD) of the blank. Technical duplicates were performed for mAbs and technical triplicates were performed for sera samples.

### ELISA assay with RBD, NTD and S2 subunits

Identification of mAbs and plasma screening of vaccinees against RBD, NTD or S2 SARS-CoV-2 protein were performed by ELISA. In brief, 3 μg/ml of RBD, NTD or S2 SARS-CoV-2 protein diluted in carbonate-bicarbonate buffer (E107, Bethyl Laboratories) was coated in 384-well plates (microplate clear, Greiner Bio-one). After overnight incubation at 4 °C, plates were washed three times with washing buffer (phosphate-buffered saline and 0.05% Tween-20) and blocked with 50 μl per well of blocking buffer (phosphate-buffered saline and 1% BSA) for 1 h at 37 °C. After washing, plates were incubated 1 h at 37 °C with mAbs diluted 1:5 in sample buffer (phosphate-buffered saline, 1% BSA and 0.05% Tween-20) or with plasma at a starting dilution of 1:10 and step diluted of 1:2 in sample buffer. Wells with no sample added were consider blank controls. Anti-human IgG–peroxidase antibody (Fab specific) produced in goat (Sigma) diluted 1:45,000 in sample buffer was then added and samples were incubated for 1 h at 37 °C. Plates were then washed, incubated with TMB substrate (Sigma) for 15 min before adding the stop solution ($H_2SO_4$ 0.2 M). The OD values were identified using the Varioskan Lux Reader (Thermo Fisher Scientific) at 450 nm. Each condition was tested in triplicate and samples tested were considered positive if the OD value was twofold the blank.

### Flow cytometry-based competition assay

To classify mAb candidates on the basis of their interaction with S epitopes, we performed a flow cytometry-based competition assay. In detail, magnetic beads (Dynabeads His-Tag, Invitrogen) were coated with histidine-tagged S protein according to the manufacturer's instructions. Then, 20 μg/ml of coated S protein beads were pre-incubated with unlabelled nAb candidates diluted 1:2 in PBS for 40 min at room temperature. After incubation, the mix beads–antibodies was washed with 100 μl of 1% PBS-BSA. Then, to analyse epitope competition, mAbs that are able to bind to the RBD (J08 and S309), NTD (4A8) or S2 domain (L19) of the S protein were labelled with four different fluorophores (Alexa Fluor 647, 488, 594 and 405) using the Alexa Fluor NHS Ester kit (Thermo Scientific), were mixed and incubated with S protein beads. Following 40 min of incubation at room temperature, the mix beads–antibodies was washed with PBS, resuspended in 150 μl of 1% PBS-BSA and analysed using the BD LSR II flow cytometer (Becton Dickinson). Beads with or without S protein incubated with labelled antibodies mix were used as positive and negative controls, respectively. FACSDiva Software (version 9) was used for data acquisition and analysis was performed using FlowJo (version 10).

### SARS-CoV-2 authentic viruses neutralization assay

All SARS-CoV-2 authentic virus neutralization assays were performed in the biosafety level 3 (BSL3) laboratories at Toscana Life Sciences in Siena (Italy) and Vismederi Srl, Siena (Italy). BSL3 laboratories are approved by a certified biosafety professional and are inspected every year by local authorities. To evaluate the neutralization activity of identified nAbs against SARS-CoV-2 and all VoCs and to evaluate the breadth of neutralization of this antibody, CPE-MN was performed[5]. In brief, for the CPE-based neutralization assay, we co-incubated J08 with a SARS-CoV-2 viral solution containing 100 median tissue culture infectious dose (100 $TCID_{50}$) of virus, and after 1 h of incubation at 37 °C, 5% $CO_2$. The mixture was then added to the wells of a 96-well plate containing a sub-confluent Vero E6 cell monolayer. Plates were incubated for 3–4 days at 37 °C in a humidified environment with 5% $CO_2$, then examined for CPE by means of an inverted optical microscope by two independent operators. All nAbs were tested at a starting dilution of 1:5 and the $IC_{100}$ was evaluated based on their initial concentration, while plasma samples were tested starting at a 1:10 dilution. Both nAbs and plasma samples were then step diluted 1:2. Technical duplicates were performed for both nAbs and plasma samples. In each plate, positive and negative controls were used as previously described[5].

### SARS-CoV-2 virus variants CPE-MN neutralization assay

The SARS-CoV-2 viruses used to perform the CPE-MN neutralization assay were the original SARS-CoV-2 virus first detected in Wuhan

(SARS-CoV-2/INMI1-Isolate/2020/Italy: MT066156), SARS-CoV-2 B.1.1.7 (INMI GISAID accession number: EPI_ISL_736997), SARS-CoV-2 B.1.351 (EVAg Cod: 014V-04058), B.1.1.248 (EVAg CoD: 014V-04089) and B.1.617.2 (GISAID ID: EPI_ISL_2029113)[26].

## Single-cell RT–PCR and Ig gene amplification and transcriptionally active PCR expression

The whole process for nAbs heavy and light chain recovery, amplification and transcriptionally active PCR (TAP) expression was performed as previously described[5]. In brief, 5 µl of cell lysate was mixed with 1 µl of random hexamer primers (50 ng/µl), 1 µl of dNTP-Mix (10 mM), 2 µl of 0.1 M DTT, 40 U/µl of RNAse OUT, $MgCl_2$ (25 mM), 5× FS buffer and Superscript IV reverse transcriptase (Invitrogen) to perform RT–PCR. Reverse transcription (RT) reaction was performed at 42 °C for 10 min, 25 °C for 10 min, 50 °C for 60 min and 94 °C for 5 min. Two rounds of PCR were performed to obtain the heavy (VH) and light (VL) chain amplicons. All PCRs were performed in a nuclease-free water (DEPC) in a total volume of 25 µl per well. For PCR I, 4 µl of cDNA were mixed with 10 µM of VH and 10 µM of VL primer-mix, 10 mM of dNTP mix, 0.125 µl of Kapa Long Range Polymerase (Sigma), 1.5 µl of MgCl2 and 5 µl of 5× Kapa Long Range Buffer. The PCR I reaction was performed at 95 °C for 3 min, 5 cycles at 95 °C for 30 s, 57 °C for 30 s, 72 °C for 30 s and 30 cycles at 95 °C for 30 s, 60 °C for 30 s, 72 °C for 30 s and a final extension of 72 °C for 2 min. Nested PCR II was performed as above starting from 3.5 µl of unpurified PCR I product. PCR II products were purified by the Millipore MultiScreen PCRµ96 plate according to manufacturer's instructions and eluted in 30 µl of nuclease-free water (DEPC). As for TAP expression, vectors were initially digested using restriction enzymes AgeI, SalI and Xho as previously described and PCR II products ligated by using the Gibson Assembly NEB into 25 ng of respective human Igγ1, Igκ and Igλ expression vectors[27,28]. TAP reaction was performed using 5 µl of Q5 polymerase (NEB), 5 µl of GC Enhancer (NEB), 5 µl of 5X buffer, 10 mM of dNTPs, 0.125 µl of forward/reverse primers and 3 µl of ligation product, using the following cycles: 98 °C for 2 min, 35 cycles 98 °C for 10 s, 61 °C for 20 s, 72 °C for 1 min and 72 °C for 5 min. TAP products were purified under the same PCR II conditions, quantified by the Qubit Fluorometric Quantitation assay (Invitrogen) and used for transient transfection in the Expi293F cell line following the manufacturer's instructions.

## Functional repertoire analyses

The VH and VL sequence reads of nAbs were manually curated and retrieved using CLC sequence viewer (Qiagen). Aberrant sequences were removed from the dataset. Analysed reads were saved in FASTA format and the repertoire analyses were performed using Cloanalyst (http://www.bu.edu/computationalimmunology/research/software/)[29,30].

## Statistical analysis

Statistical analysis was assessed with GraphPad Prism Version 8.0.2 (GraphPad Software). Non-parametric Mann–Whitney $t$-test was used to evaluate statistical significance between the two groups analysed in this study. Statistical significance was shown as *$P \le 0.05$, **$P \le 0.01$, ***$P \le 0.001$, ****$P \le 0.0001$.

## Reporting summary

Further information on research design is available in the Nature Research Reporting Summary linked to this paper.

## Data availability

Source data are provided with this paper. All data supporting the findings in this study are available within the article or can be obtained from the corresponding author on request. Source data are provided with this paper.

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

**Acknowledgements** This work was funded by the European Research Council (ERC) advanced grant agreement number 787552 (vAMRes). This publication was supported by funds from the 'Centro Regionale Medicina di Precisione' and by all of the people who answered the call to fight with us in the battle against SARS-CoV-2 with their kind donations on the platform ForFunding (https://www.forfunding.intesasanpaolo.com/DonationPlatform-ISP/nav/progetto/id/3380). This work was funded by COOP ITALIA Soc. Coop. This publication was supported by the European Virus Archive goes Global (EVAg) project, which has received funding from the European Union's Horizon 2020 research and innovation programme under grant agreement number 653316. This publication was supported by the COVID-2020-12371817 project, which has received funding from the Italian Ministry of Health. We also acknowledge J. McLellan, for kindly providing the S protein trimer, RBD, NTD and S2 constructs; O. Schwartz, for providing the B.1.617.2 (Delta) SARS-CoV-2 variant; the nursing staff of the operative unit of the department of Medical Sciences, Infectious and Tropical Diseases Unit, Siena University Hospital, Siena, Italy; and all of the donors who are vaccinated against COVID-19 for participating to this study.

**Author contributions** E.A. and R.R. conceived the study. F.M., M.F., I.R. and M.T. enrolled the COVID-19 vaccinees. E.A. and I.P. performed PBMC isolation and single-cell sorting. I.P. performed ELISAs and competition assays. I.P. and N.M. recovered nAbs expressing VH and VL and antibodies. P.P. and E.A. recovered the VH and VL sequences and performed the repertoire analyses. E.P. and V.A. produced and purified the SARS-CoV-2 S protein constructs. E.A., G.P., I.H., M.L., L.B. and G.G. performed the neutralization assays in the BSL3 facilities. C.D.S. supported day-to-day laboratory activities and management. E.A. and R.R. wrote the manuscript. E.A., I.P., G.P., N.M., P.P., I.H., M.L., E.P., V.A., L.B., G.G., C.D.S., M.F., I.R., M.T., F.M., C.S., E.M. and R.R. undertook the final revision of the manuscript. E.A., C.S., E.M. and R.R. coordinated the project.

**Competing interests** R.R. is an employee of the GSK group of companies. E.A., I.P., N.M., P.P., E.P., C.D.S., C.S. and R.R. are listed as inventors of full-length human mAbs described in Italian patent applications no. 102020000015754 filed on 30 June 2020, 102020000018955 filed on 3 August 2020 and 102020000029969 filed on 4 December 2020, and the international patent system number PCT/IB2021/055755 filed on 28 June 2021. All patents were submitted by Fondazione Toscana Life Sciences, Siena, Italy. The remaining authors declare no competing interests.

**Additional information**
**Correspondence and requests for materials** should be addressed to Rino Rappuoli.

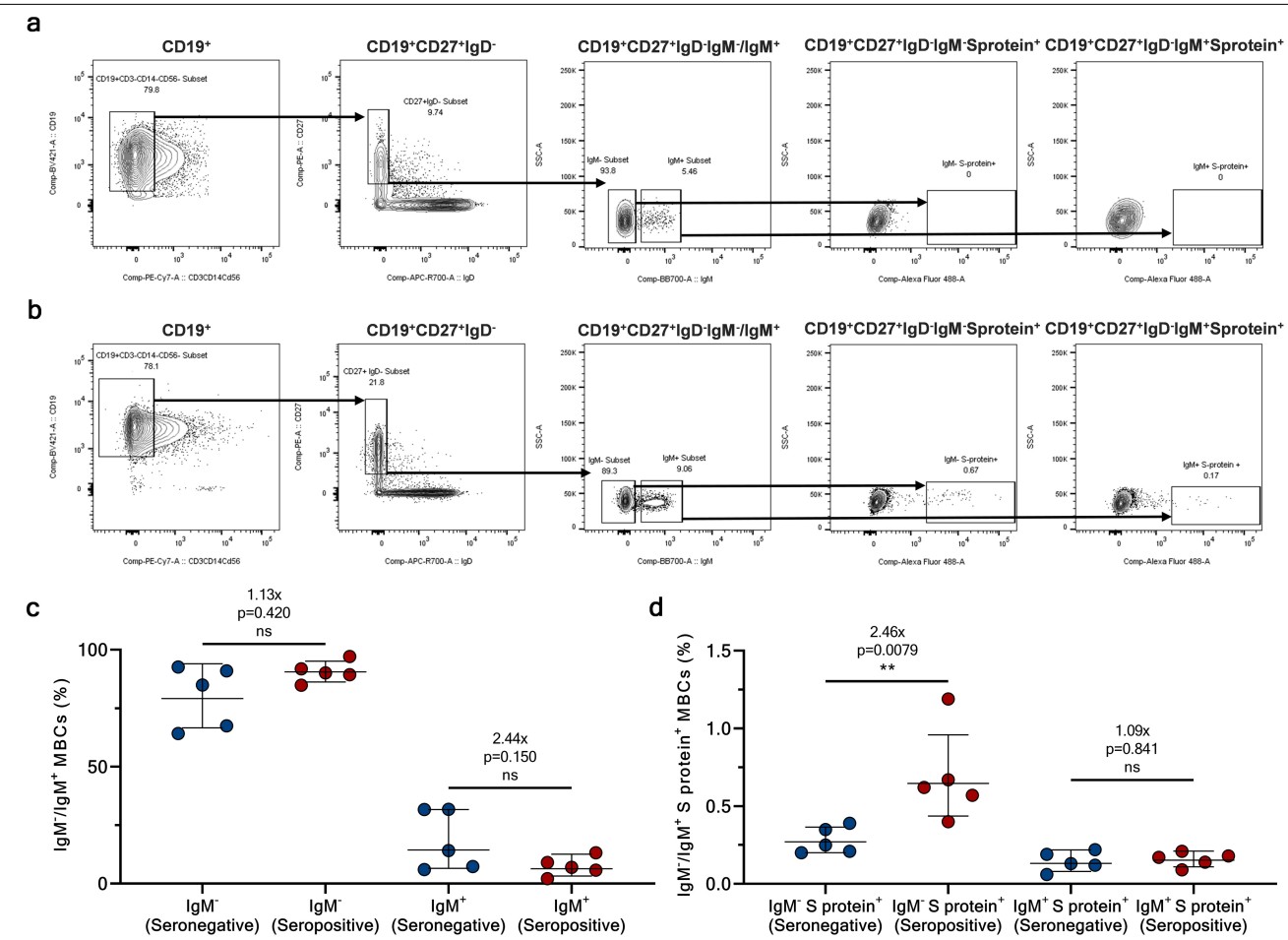

| Subject | SARS-CoV-2 serology | IgM⁻ Subset (%) | IgM⁻ S protein⁺ Subset (%) | IgM⁺ Subset (%) | IgM⁺ S protein⁺ Subset (%) |
|---|---|---|---|---|---|
| VAC-001 | Seronegative | 67.50 | 0.21 | 31.70 | 0.13 |
| VAC-002 | Seronegative | 64.20 | 0.25 | 31.80 | 0.06 |
| VAC-007 | Seronegative | 92.60 | 0.35 | 6.02 | 0.22 |
| VAC-008 | Seronegative | 84.90 | 0.39 | 14.10 | 0.12 |
| VAC-010 | Seronegative | 91.00 | 0.20 | 7.27 | 0.19 |
| VAC-003 | Seropositive | 91.80 | 0.57 | 5.77 | 0.21 |
| VAC-004 | Seropositive | 97.10 | 0.62 | 2.14 | 0.14 |
| VAC-005 | Seropositive | 89.30 | 0.67 | 9.06 | 0.17 |
| VAC-006 | Seropositive | 90.10 | 0.40 | 6.94 | 0.18 |
| VAC-009 | Seropositive | 84.80 | 1.19 | 13.20 | 0.09 |

**Extended Data Fig. 1 | Single cell sorting and memory B cell frequencies.** **a**, **b**, The gating strategy shows from left to right: CD19⁺ B cells; CD19⁺CD27⁺IgD⁻; CD19⁺CD27⁺IgD⁻IgM⁻/IgM⁺; CD19⁺CD27⁺IgD⁻IgM⁻Sprotein⁺; CD19⁺CD27⁺IgD⁻IgM⁺Sprotein⁺ for a healthy donor (used as negative control for S protein staining) and a vaccinated subject. **c**, The graph shows the frequency of CD19⁺CD27⁺IgD⁻IgM⁻ and IgM⁺ in seronegative (n=5) and seropositive donors (n=5). **d**, The graph shows the frequency of CD19⁺CD27⁺IgD⁻IgM⁻ and IgM⁺ able to bind the SARS-CoV-2 S protein trimer (S protein⁺) in seronegative (n=5) and seropositive (n=5) donors. Geometric mean and standard deviation are denoted on the graphs. A nonparametric Mann–Whitney t test was used to evaluate statistical significances between groups. Two-tailed p-value significances are shown as *p < 0.05, **p < 0.01, ***p < 0.001, and ****p < 0.0001. **e**, The table summarizes the frequencies of the cell population above described for all subjects enrolled in our study.

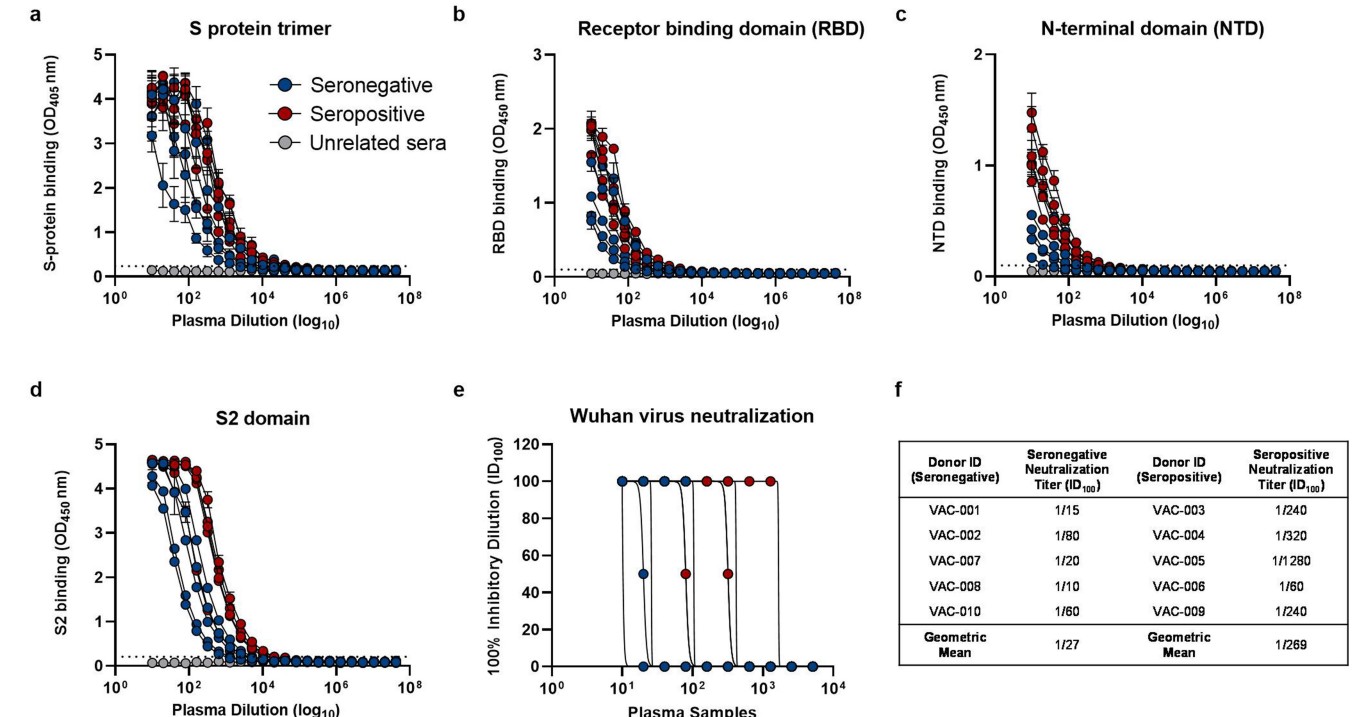

**Extended Data Fig. 2 | Plasma response of COVID-19 vaccinees. a–d**, Graphs show the ability of plasma samples from seronegative and seropositive vaccinees to bind the S protein trimer, RBD, NTD and S2 domain. Mean and standard deviation are denoted on each graph. Technical triplicates were performed for each experiment. **e**, The graph shows the neutralizing activity of plasma samples against the original Wuhan SARS-CoV-2 virus. Technical duplicates were performed for each experiment. **f**, The table summarizes the 100% inhibitory dilution ($ID_{100}$) of each COVID-19 vaccinee and the geometric mean for seronegative and seropositive donors.

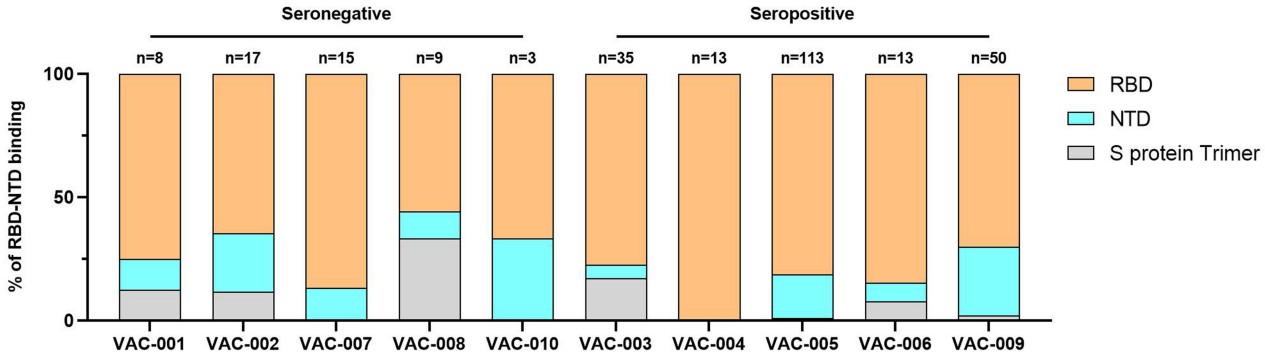

**Extended Data Fig. 3 | RBD and NTD binding distribution of nAbs.** The graph shows the percentage of antibodies that bind specifically the RBD (light orange) or the NTD (cyan) or that did not bind single domains but recognized exclusively the S protein in its trimetric conformation (gray). The number (n) of tested nAbs per donor is reported on top of each bar. Technical duplicates were performed for each experiment.

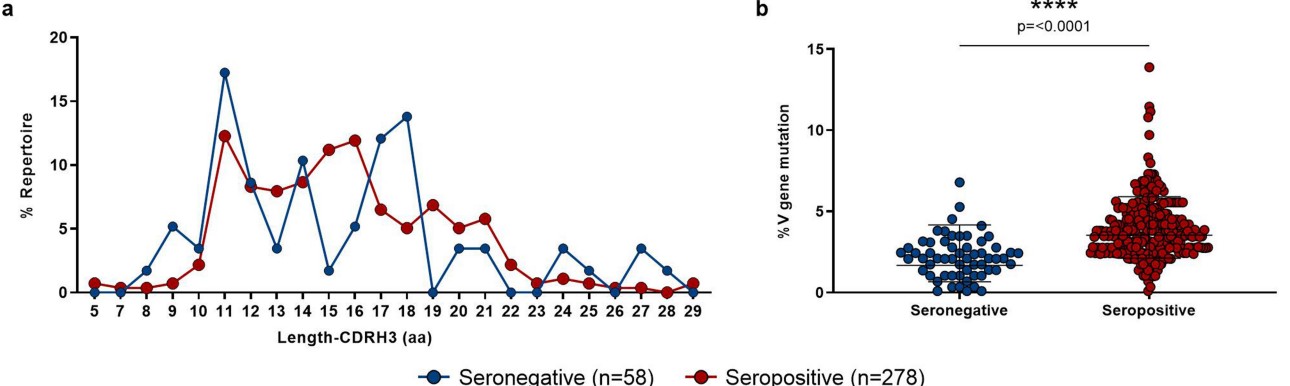

**Extended Data Fig. 4 | Heavy chain CDR3 length and somatic hypermutation levels in seronegative and seropositive vaccinees. a**, The graph shows the heavy chain CDR3 length represented in amino acids (aa). **b**, The graph shows the overall somatic hypermutation level of nAbs isolated from seronegative and seropositive vaccinees. Geometric mean and standard deviation are denoted on the graphs. A nonparametric Mann–Whitney t test was used to evaluate statistical significances between groups. Two-tailed p-value significances are shown as *p < 0.05, **p < 0.01, ***p < 0.001, and ****p < 0.0001.

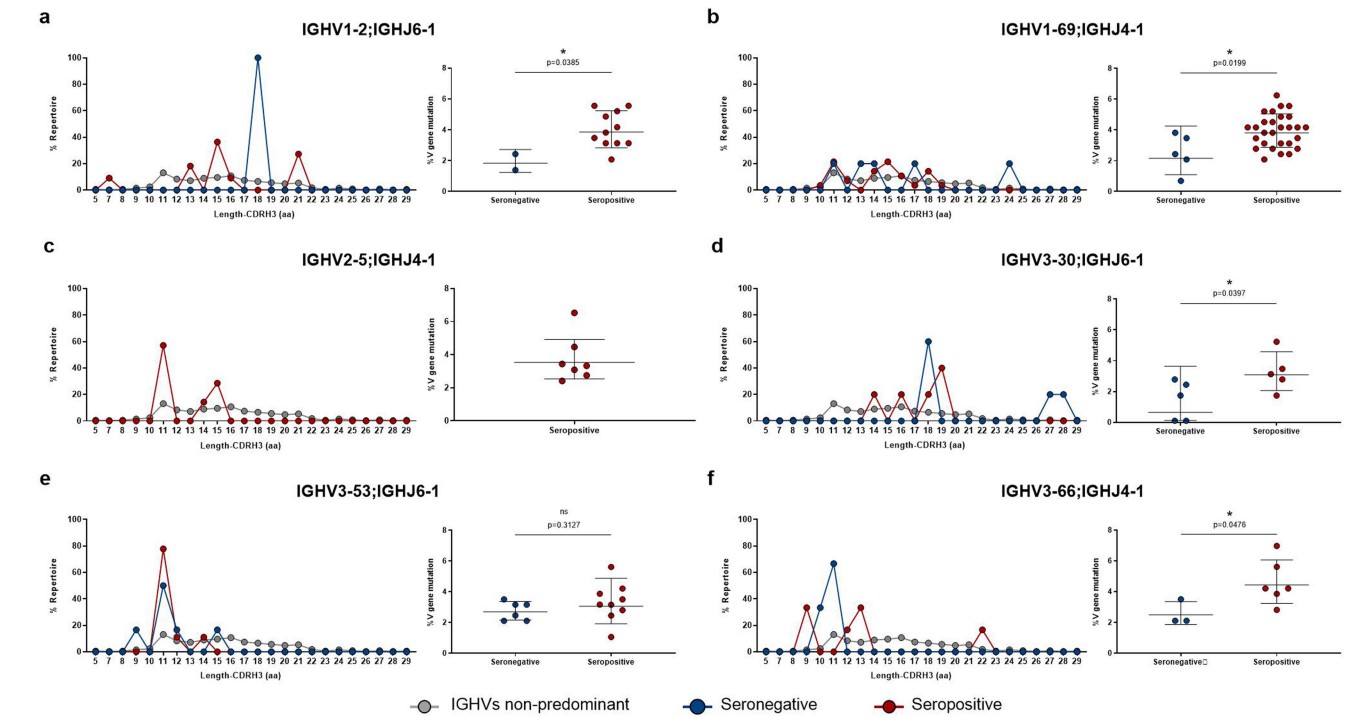

**Extended Data Fig. 5 | Heavy chain CDR3 length and somatic hypermutation levels of predominant gene derived nAbs. a–f**, Graphs show the amino acidic heavy chain CDR3 length (left panel) and the somatic hypermutation level (right panel) of nAbs derived from the IGHV1-2;IGHJ6-1 (*n* = 13), IGHV1-69;IGHJ4-1 (*n* = 33), IGHV2-5;IGHJ4-1 (*n* = 7), IGHV3-30;IGHJ6-1 (*n* = 10), IGHV3-53;IGHJ6-1 (*n* = 15) and IGHV3-66;IGHJ4-1 (*n* = 9) gene families. Geometric mean and standard deviation are denoted on the graphs. A nonparametric Mann–Whitney t test was used to evaluate statistical significances between groups. Two-tailed p-value significances are shown as \**p* < 0.05, \*\**p* < 0.01, \*\*\**p* < 0.001, and \*\*\*\**p* < 0.0001.

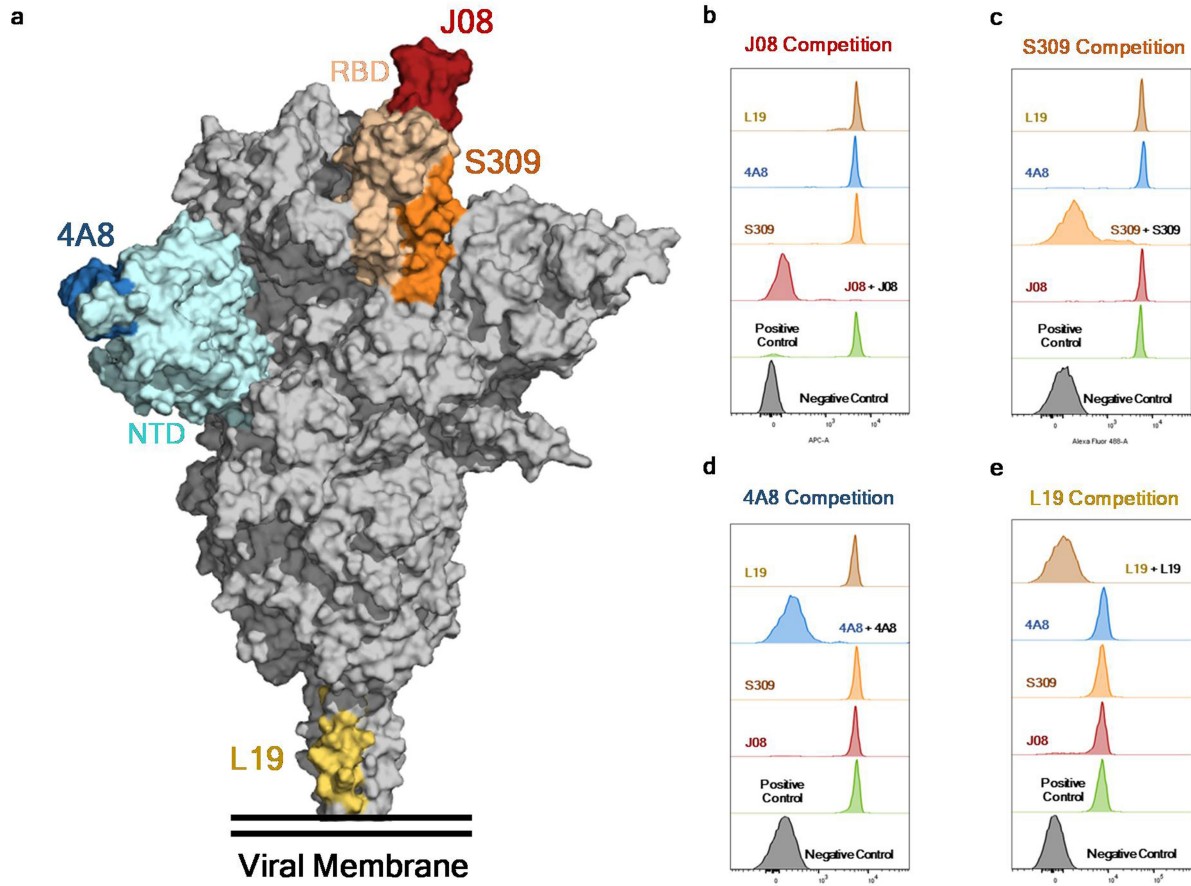

**Extended Data Fig. 6 | Epitope binning assay. a**, Schematic representation of the epitopes recognized by J08 (dark red), S309 (orange), 4A8 (dark blue) and L19 (gold), mAbs on the S protein surface. **b**–**e**, Representative cytometer peaks per each of the four mAbs used for the competition assay. Positive (beads conjugated with only primary labeled antibody) and negative (un-conjugated beads) controls are shown as green and gray peaks, respectively.

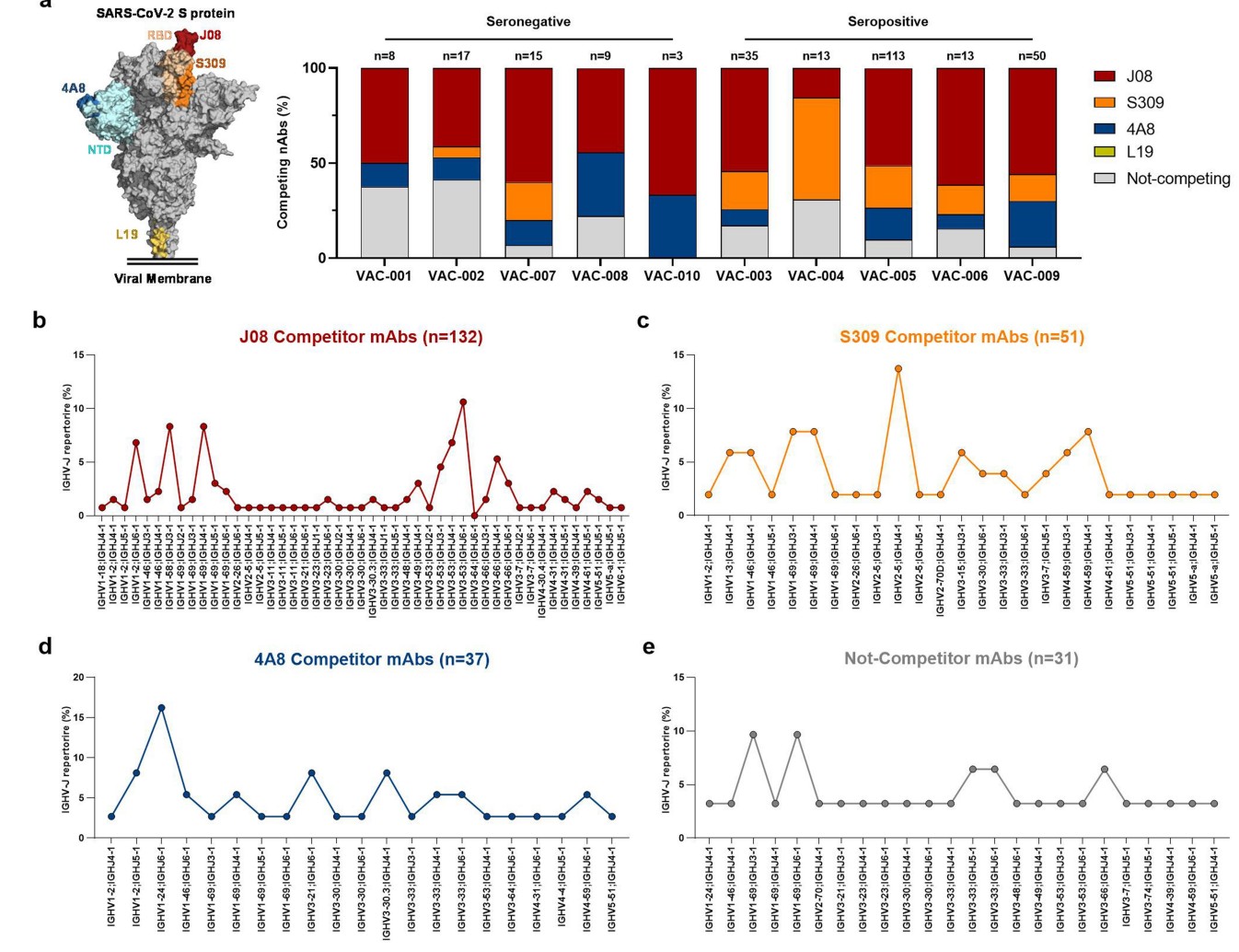

**Extended Data Fig. 7 | Epitope binning and genetic characterization of competing nAbs. a**, The bar graph shows the percentage (%) of nAbs competing with J08 (dark red), S309 (orange), 4A8 (dark blue) and L19 (gold), or antibodies that did not compete with any of the previous mAbs (gray). A schematic representation of J08, S309, 4A8 and L19 epitopes on the S protein surface is shown on the left side of the panel. **b**–**e**, Graphs show the IGHV-J rearrangement percentage for nAbs that competed against J08, S309, 4A8, or that did not compete with any of these mAbs. The total number (n) of competing nAbs per group is shown on top of each graph.

**Extended Data Table 1 | Clinical details of COVID-19 vaccinees**

| Subject ID | Gender | Age | Previous COVID-19 negative test | COVID-19 positive test | Type of test | Severity of Infection | SARS-CoV-2 Serology | First Dose (dd/mm/yy) | Second Dose (dd/mm/yy) | Blood Collection (dd/mm/yy) | Days from Infection to First Dose | Days from Last Dose to Blood Collection |
|---|---|---|---|---|---|---|---|---|---|---|---|---|
| VAC-001 | M | 38 | 16/12/2020 | Not-applicable | Swab | Not-applicable | Seronegative | 27/12/2020 | 18/01/2021 | 02/03/2021 | Not-applicable | 43 |
| VAC-002 | F | 38 | 31/12/2020 | Not-applicable | Swab | Not-applicable | Seronegative | 01/01/2021 | 22/01/2021 | 02/03/2021 | Not-applicable | 39 |
| VAC-007 | M | 38 | 29/12/2020 | Not-applicable | Swab | Not-applicable | Seronegative | 04/01/2021 | 25/01/2021 | 31/03/2021 | Not-applicable | 65 |
| VAC-008 | M | 43 | 28/12/2020 | Not-applicable | Swab | Not-applicable | Seronegative | 03/01/2021 | 24/01/2021 | 31/03/2021 | Not-applicable | 66 |
| VAC-010 | F | 51 | 15/02/2021 | Not-applicable | Swab | Not-applicable | Seronegative | 18/02/2021 | 11/03/2021 | 07/04/2021 | Not-applicable | 27 |
| VAC-003 | M | 38 | 15/10/2020 | 26/10/2020 | Swab | Asymptomatic | Seropositive | 08/01/2021 | 15/02/2021 | 09/03/2021 | 74 | 22 |
| VAC-004 | F | 25 | Not-applicable | 22/10/2020 | Swab | Mild | Seropositive | 08/02/2021 | 01/03/2021 | 09/03/2021 | 78 | 8 |
| VAC-005 | M | 25 | 01/08/2020 | 02/11/2020 | Serological | Mild | Seropositive | 11/01/2021 | 16/02/2021 | 16/03/2021 | 71 | 28 |
| VAC-006 | F | 57 | 27/04/2020 | 24/10/2020 | Swab | Asymptomatic | Seropositive | 16/01/2021 | 11/02/2021 | 16/03/2021 | 79 | 33 |
| VAC-009 | M | 46 | 29/09/2020 | 06/11/2020 | Serological | Moderate | Seropositive | 20/03/2021 | Not-applicable | 07/04/2021 | 134 | 18 |

**Extended Data Table 2 | Summary of B cell frequencies and antibodies of COVID-19 vaccinees**

| Subject | SARS-CoV-2 serology | S protein+ MBCs Sorted | S protein+ mAbs (n) | S protein+ mAbs (%) | Wuhan - Neutralizing antibodies (n) | Wuhan - Neutralizing antibodies (%) | B.1.1.7 - Neutralizing antibodies (n) | B.1.1.7 - Neutralizing antibodies (%) | B.1.351 - Neutralizing antibodies (n) | B.1.351 - Neutralizing antibodies (%) | B.1.1.248 - Neutralizing antibodies (n) | B.1.1.248 - Neutralizing antibodies (%) |
|---|---|---|---|---|---|---|---|---|---|---|---|---|
| VAC-001 | Seronegative | 484 | 205 | 42.3 | 13 | 6.3 | 8 | 61.5 | 2 | 15.4 | 6 | 46.2 |
| VAC-002 | Seronegative | 497 | 316 | 63.6 | 27 | 8.5 | 19 | 70.4 | 6 | 22.2 | 11 | 40.7 |
| VAC-007 | Seronegative | 161 | 100 | 62.1 | 16 | 16.0 | 14 | 87.5 | 3 | 18.8 | 6 | 37.5 |
| VAC-008 | Seronegative | 682 | 107 | 15.7 | 10 | 9.3 | 6 | 60.0 | 5 | 50.0 | 6 | 60.0 |
| VAC-010 | Seronegative | 528 | 216 | 40.9 | 5 | 2.3 | 3 | 60.0 | 0 | 0.0 | 2 | 40.0 |
| Total (Seronegative) | | 2,352 | 944 | 40.1 | 71 | 7.5 | 50 | 70.4 | 16 | 22.5 | 31 | 43.6 |
| VAC-003 | Seropositive | 616 | 428 | 69.5 | 78 | 18.2 | 58 | 74.4 | 25 | 32.1 | 40 | 51.3 |
| VAC-004 | Seropositive | 506 | 318 | 62.8 | 31 | 9.7 | 24 | 77.4 | 11 | 35.5 | 18 | 58.1 |
| VAC-005 | Seropositive | 924 | 601 | 65.0 | 152 | 25.3 | 132 | 86.8 | 78 | 51.3 | 114 | 75.0 |
| VAC-006 | Seropositive | 396 | 216 | 55.8 | 19 | 8.8 | 12 | 63.2 | 5 | 26.3 | 8 | 42.1 |
| VAC-009 | Seropositive | 1,090 | 736 | 67.5 | 60 | 8.1 | 48 | 80.0 | 15 | 25.0 | 31 | 51.7 |
| Total (Seropositive) | | 3,532 | 2,299 | 65.1 | 340 | 14.8 | 274 | 80.6 | 134 | 39.4 | 211 | 62.0 |

## Extended Data Table 3 | Competition assay summary

| Subject | SARS-CoV-2 serology | Distribution Competition J08 (n) | Distribution Competition J08 (%) | Distribution Competition S309 (n) | Distribution Competition S309 (%) | Distribution Competition 4A8 (n) | Distribution Competition 4A8 (%) | Distribution Competition L19 (n) | Distribution Competition L19 (%) | Distribution Competition Not-competing (n) | Distribution Competition Not-competing (%) |
|---|---|---|---|---|---|---|---|---|---|---|---|
| VAC-001 | Seronegative | 4 | 50.0 | 0 | 0.0 | 1 | 12.5 | 0 | 0.0 | 3 | 37.5 |
| VAC-002 | Seronegative | 7 | 41.2 | 1 | 5.9 | 2 | 11.8 | 0 | 0.0 | 7 | 41.2 |
| VAC-007 | Seronegative | 9 | 60.0 | 3 | 20.0 | 2 | 13.3 | 0 | 0.0 | 1 | 6.7 |
| VAC-008 | Seronegative | 4 | 44.4 | 0 | 0.0 | 3 | 33.3 | 0 | 0.0 | 2 | 22.2 |
| VAC-010 | Seronegative | 2 | 66.7 | 0 | 0.0 | 1 | 33.3 | 0 | 0.0 | 0 | 0.0 |
| Total (Seronegative) | | 26 | 50.0 | 4 | 7.7 | 9 | 17.3 | 0 | 0.0 | 13 | 25.0 |
| VAC-003 | Seropositive | 19 | 54.3 | 7 | 20.0 | 3 | 8.6 | 0 | 0.0 | 6 | 17.1 |
| VAC-004 | Seropositive | 2 | 15.4 | 7 | 53.8 | 0 | 0.0 | 0 | 0.0 | 4 | 30.8 |
| VAC-005 | Seropositive | 58 | 51.3 | 25 | 22.1 | 19 | 16.8 | 0 | 0.0 | 11 | 9.7 |
| VAC-006 | Seropositive | 8 | 61.5 | 2 | 15.4 | 1 | 7.7 | 0 | 0.0 | 2 | 15.4 |
| VAC-009 | Seropositive | 28 | 56.0 | 7 | 14.0 | 12 | 24.0 | 0 | 0.0 | 3 | 6.0 |
| Total (Seropositive) | | 115 | 51.3 | 48 | 21.4 | 35 | 15.6 | 0 | 0.0 | 26 | 11.6 |

# Reporting Summary

## Statistics

For all statistical analyses, confirm that the following items are present in the figure legend, table legend, main text, or Methods section.

| n/a | Confirmed | |
|---|---|---|
| ☐ | ☒ | The exact sample size ($n$) for each experimental group/condition, given as a discrete number and unit of measurement |
| ☐ | ☒ | A statement on whether measurements were taken from distinct samples or whether the same sample was measured repeatedly |
| ☐ | ☒ | The statistical test(s) used AND whether they are one- or two-sided<br>*Only common tests should be described solely by name; describe more complex techniques in the Methods section.* |
| ☒ | ☐ | A description of all covariates tested |
| ☒ | ☐ | A description of any assumptions or corrections, such as tests of normality and adjustment for multiple comparisons |
| ☐ | ☒ | A full description of the statistical parameters including central tendency (e.g. means) or other basic estimates (e.g. regression coefficient) AND variation (e.g. standard deviation) or associated estimates of uncertainty (e.g. confidence intervals) |
| ☐ | ☒ | For null hypothesis testing, the test statistic (e.g. $F$, $t$, $r$) with confidence intervals, effect sizes, degrees of freedom and $P$ value noted<br>*Give P values as exact values whenever suitable.* |
| ☒ | ☐ | For Bayesian analysis, information on the choice of priors and Markov chain Monte Carlo settings |
| ☒ | ☐ | For hierarchical and complex designs, identification of the appropriate level for tests and full reporting of outcomes |
| ☒ | ☐ | Estimates of effect sizes (e.g. Cohen's $d$, Pearson's $r$), indicating how they were calculated |

*Our web collection on statistics for biologists contains articles on many of the points above.*

## Software and code

Policy information about availability of computer code

| Data collection | - Thermo Fisher SkanIt Software Microplate Readers 6.0.1<br>- BD Biosciences BD FACSDiva Software v9.0 |
|---|---|
| Data analysis | - GraphPad Prism 8.0.2 was used to perform statistical analyses<br>- BD FlowJo 10.5.3<br>- Qiagen CLC sequence viewer 350 8.0.0<br>- Boston University, Cloanalyst (http://www.bu.edu/computationalimmunology/research/software/) |

For manuscripts utilizing custom algorithms or software that are central to the research but not yet described in published literature, software must be made available to editors and reviewers. We strongly encourage code deposition in a community repository (e.g. GitHub). See the Nature Portfolio guidelines for submitting code & software for further information.

## Data

Policy information about availability of data

All manuscripts must include a data availability statement. This statement should provide the following information, where applicable:

- Accession codes, unique identifiers, or web links for publicly available datasets
- A description of any restrictions on data availability
- For clinical datasets or third party data, please ensure that the statement adheres to our policy

Source data are provided with this paper. All data supporting the findings in this study are available within the article or can be obtained from the corresponding author upon request.

# Field-specific reporting

Please select the one below that is the best fit for your research. If you are not sure, read the appropriate sections before making your selection.

☒ Life sciences ☐ Behavioural & social sciences ☐ Ecological, evolutionary & environmental sciences

For a reference copy of the document with all sections, see nature.com/documents/nr-reporting-summary-flat.pdf

# Life sciences study design

All studies must disclose on these points even when the disclosure is negative.

| Sample size | 10 subjects in total, 5 seronegative and 5 seropositive, were analyzed in this study. A total of 2,352 and 3,532 spike protein specific memory B cells from seronegative and seropositive subjects were tested in this study. Given the exploratory nature of the study, we did not use statistical methods to predetermine sample size. Sample size was based on previous studies that applied a similar technology. The authors believed that 5 subjects/group were a good balance between feasibility of analyzing at single cell level several thousands of memory B cells and the ability to represent the antibody response of seronegative and seropositve people. |
|---|---|
| Data exclusions | No data was excluded. |
| Replication | All experiments were performed in technical duplicates or triplicates as indicated in the figure legends and methods section. |
| Randomization | The experiments were not randomized and all available samples were tested. The authors aimed to specifically assess the antibody response of seronegative and seropositve subjects. Donors were specifically recruited based on their previous infection and vaccination history. Randomization would have not allowed to enroll 5 subjects/group which was our technical limit for single cell analysis of the antibody response. Based on what stated above, the authors believed that randomization was not appropriate. |
| Blinding | The investigators were not blinded during group allocation, data collection and analyses. The clinical protocol established to enroll subjects in this study reports information regarding previous infection and vaccination in order to allocate 5 subjects/group. Pseudonymized information received in the lab reports the same information and therefore blinding for group allocation was not possible. |

# Reporting for specific materials, systems and methods

We require information from authors about some types of materials, experimental systems and methods used in many studies. Here, indicate whether each material, system or method listed is relevant to your study. If you are not sure if a list item applies to your research, read the appropriate section before selecting a response.

## Materials & experimental systems

| n/a | Involved in the study |
|---|---|
| ☐ | ☒ Antibodies |
| ☐ | ☒ Eukaryotic cell lines |
| ☒ | ☐ Palaeontology and archaeology |
| ☒ | ☐ Animals and other organisms |
| ☐ | ☒ Human research participants |
| ☒ | ☐ Clinical data |
| ☒ | ☐ Dual use research of concern |

## Methods

| n/a | Involved in the study |
|---|---|
| ☒ | ☐ ChIP-seq |
| ☐ | ☒ Flow cytometry |
| ☒ | ☐ MRI-based neuroimaging |

# Antibodies

| Antibodies used | BD Biosciences CD19 BV421, Cat#562440, Clone ID HIB19, Lot#8270584<br>BD Biosciences IgM PerCP-Cy5.5, Cat#561285, Clone ID G20-127, Lot#9269055<br>BD Biosciences CD27 PE, Cat#340425, Clone ID L128, Lot#9288842<br>BD Biosciences IgD-A700, Cat#561302, Clone ID IA6-2, Lot#9199226<br>BioLegend CD3 PE-Cy7, Cat#300420, Clone ID UCHT1, Lot#B303315<br>BioLegend CD14 PE-Cy7, Cat#301814, Clone ID M5E2, Lot#B272337<br>BioLegend CD56 PE-Cy7, Cat#318318, Clone ID HCD56, Lot#B297987<br>Southern Biotech Goat Anti-Human IgG-Alkaline Phosphatase, Cat#2040-04, polyclonal, Lot#K2119-XG00B<br>Southern Biotech Goat Anti-Human IgA-Alkaline Phosphatase, Cat#2050-04, polyclonal, Lot#G0919-W620C<br>Sigma-Aldrich Anti-Human IgG (Fab specific)–Peroxidase antibody produced in goat, Cat#A0293, polyclonal, Lot#019M4876V |
|---|---|
| Validation | BD Biosciences CD19 BV421, Cat#562440, Clone ID HIB19, QC testing, reactivity human, application flow cytometry (https://www.bdbiosciences.com/content/bdb/paths/generate-tds-document.us.562440.pdf).<br>BD Biosciences IgM PerCP-Cy5.5, Cat#561285, Clone ID G20-127, QC testing, reactivity human, application flow cytometry (https://www.bdbiosciences.com/content/bdb/paths/generate-tds-document.us.561285.pdf).<br>BD Biosciences CD27 PE, Cat#340425, Clone ID L128, QC testing, reactivity human, application flow cytometry (https:// |

www.bdbiosciences.com/en-us/products/reagents/flow-cytometry-reagents/clinical-discovery-research/single-color-antibodies-ruo-gmp/pe-mouse-anti-human-cd27.340425)
BD Biosciences IgD-A700, Cat#561302, Clone ID IA6-2, QC testing, reactivity human, application flow cytometry (https://www.bdbiosciences.com/content/bdb/paths/generate-tds-document.us.561302.pdf)
BioLegend CD3 PE-Cy7, Cat#300420, Clone ID UCHT1, reactivity human and cross-reactivity with chimpanzee, application flow cytometry (https://www.biolegend.com/en-us/global-elements/pdf-popup/pe-cyanine7-anti-human-cd3-antibody-3070?filename=PECyanine7%20anti-human%20CD3%20Antibody.pdf&pdfgen=true)
BioLegend CD14 PE-Cy7, Cat#301814, Clone ID M5E2, Reactivity Human, African Green, Capuchin Monkey, Cattle (Bovine, Cow), Chimpanzee, Common Marmoset, Cotton-topped Tamarin, Cynomolgus, Dog (Canine), Rhesus, Pigtailed Macaque, Squirrel Monkey, application flow cytometry (https://www.biolegend.com/en-us/global-elements/pdf-popup/pe-cyanine7-anti-human-cd14-antibody-2729?filename=PECyanine7%20anti-human%20CD14%20Antibody.pdf&pdfgen=true)
BioLegend CD56 PE-Cy7, Cat#318318, Clone ID HCD56, Reactivity Human, African Green, Baboon, Cynomolgus, Rhesus, application flow cytometry (https://www.biolegend.com/en-us/global-elements/pdf-popup/pe-cyanine7-anti-human-cd56-ncam-antibody-3802?filename=PECyanine7%20anti-human%20CD56%20NCAM%20Antibody.pdf&pdfgen=true)
Southern Biotech Goat Anti-Human IgG-Alkaline Phosphatase, Cat#2040-04, polyclonal, reactivity heavy chain of human IgG, application ELISA (https://www.southernbiotech.com/techbul/2040.pdf)
Southern Biotech Goat Anti-Human IgA-Alkaline Phosphatase, Cat#2050-04, polyclonal, reactivity heavy chain of human IgA, application ELISA (https://www.southernbiotech.com/techbul/2050.pdf)
Sigma-Aldrich Anti-Human IgG (Fab specific)–Peroxidase antibody produced in goat, Cat#A0293, polyclonal, reactivity human, application ELISA (https://www.sigmaaldrich.com/IT/en/product/sigma/a0293#)

# Eukaryotic cell lines

Policy information about cell lines

| Cell line source(s) | VERO E6 cell line ATCC Cat#CRL-1586; Expi293F cells Thermo Fisher Cat#A14527; 3T3-msCD40L Cells NIH AIDS Reagent Program Cat#12535. |
|---|---|
| Authentication | These cell lines were obtained from vendors that sell authenticated cell lines, they grew, performed and showed morphology as expected. No additional specific authentication was performed. |
| Mycoplasma contamination | Vero E6 cell lines are routinely tested on a monthly basis and tested negative for mycoplasma. 3T3-msCD40L cell line was tested negative to mycoplasma by the provider and Expi293F cells were not tested for mycoplasma contamination. |
| Commonly misidentified lines (See ICLAC register) | No commonly misidentified cell lines were used in this study. |

# Human research participants

Policy information about studies involving human research participants

| Population characteristics | This work results from a collaboration with the Azienda Ospedaliera Universitaria Senese, Siena (IT) that provided samples from COVID-19 vaccinated donors, of both sexes (4 females and 6 males), who gave their written consent. All data relevant to enrolled subjects are reported in this study. Subjects elegible for this study were of all sexes (aged 18-85) naïve or previously infected by SARS-CoV-2 and then vaccinated with the COVID-19 BNT162b2 mRNA vaccine. |
|---|---|
| Recruitment | Individuals with or without previous SARS-CoV-2 infection vaccinated with the COVID-19 BNT162b2 mRNA vaccine were enrolled by the clinicians involved in the study entitled "Isolamento di anticorpi monoclonali umani contro SARS-CoV-2 per lo sviluppo di nuove terapie e vaccini", Prot. n. TLS_SARS-CoV-2, at the Azienda Ospedaliera Universitaria Senese, Siena (IT). The authors do not see any potential bias in the generation or interpretation of the data reported in this study. |
| Ethics oversight | The study was approved by the Comitato Etico di Area Vasta Sud Est (CEAVSE) ethics committees (Parere 17065 in Siena) and conducted according to good clinical practice in accordance with the declaration of Helsinki (European Council 2001, US Code of Federal Regulations, ICH 1997). This study was unblinded and not randomized. No statistical methods were used to predetermine sample size. |

Note that full information on the approval of the study protocol must also be provided in the manuscript.

# Flow Cytometry

## Plots

Confirm that:

☒ The axis labels state the marker and fluorochrome used (e.g. CD4-FITC).

☒ The axis scales are clearly visible. Include numbers along axes only for bottom left plot of group (a 'group' is an analysis of identical markers).

☒ All plots are contour plots with outliers or pseudocolor plots.

☒ A numerical value for number of cells or percentage (with statistics) is provided.

## Methodology

| Sample preparation | Human PBMC were isolated from heparin-treated whole blood by density gradient centrifugation (Ficoll-Paque™ PREMIUM, |
|---|---|

| Sample preparation | Sigma-Aldrich). After separation, PBMC were stained with Live/Dead Fixable Aqua (Invitrogen; Thermo Scientific) diluted 1:500 at room temperature RT. After 20 min incubation cells were washed with PBS and unspecific bindings were saturated with 20% normal rabbit serum (Life technologies). Following 20 min incubation at 4°C cells were washed with PBS and stained with SARS-CoV-2 S-protein labeled with Strep-Tactin®XT DY-488 (iba-lifesciences cat# 2-1562-050) for 30 min at 4°C. After incubation the following staining mix was used CD19 V421 (BD cat# 562440, 1:320), IgM PerCP-Cy5.5 (BD cat# 561285, 1:50), CD27 PE (BD cat# 340425, 1:30), IgD-A700 (BD cat# 561302, 1:15), CD3 PE-Cy7 (BioLegend cat# 300420, 1:100), CD14 PE-Cy7 (BioLegend cat# 301814, 1:320), CD56 PE-Cy7 (BioLegend cat# 318318, 1:80) and cells were incubated at 4°C for additional 30 min. Stained MBCs were single cell-sorted with a BD FACS Aria III (BD Biosciences). |
|---|---|
| Instrument | BD FACS Aria III Cell Sorter BD Biosciences |
| Software | BD Biosciences BD FACSDiva Software v9.0 |
| Cell population abundance | Single cell sorted S protein trimer-specific (S protein+), class-switched memory B cells (CD19+CD27+IgD-IgM-) were 0.21, 0.25, 0.35, 0.39, 0.20, 0.57, 0.62, 0.67, 0.40 and 1.19% for subject VAC-001, VAC-002, VAC-007, VAC-008, VAC-010, VAC-003, VAC-004, VAC-005, VAC-006 and VAC-009 respectively. Sorted cells were gated on the CD19+CD27+IgD-IgM-S protein+ population based on the negative control as reported in the Extended Data (Extended Data Figure 1a,b). |
| Gating strategy | The gating strategy used for the single cell sorting of spike protein specific memory B cells is shown in the Extended Data (Extended Data Figure 1). Boundaries between "positive" and "negative" cells are defined and denoted on each graph. |

☒ Tick this box to confirm that a figure exemplifying the gating strategy is provided in the Supplementary Information.

