## [Peer Review File · Nature]

Manuscript Title: Hybrid immunity improves B cells and antibodies against SARS-CoV-2 variants

Reviewer Comments & Author Rebuttals

Reviewer Reports on the Initial Version:

Referee #1 (Remarks to the Author):

Andreano and colleagues compare, at the single B cell level, the antibody response to SARS-CoV-2 in donors that were infected and then vaccinated to donors that were seronegative at the time of vaccination. While others have very recently reported that infection followed by vaccination can elicit antibodies that bind to spike proteins more avidly and more broadly than vaccination alone, the authors have extended these observations to the level of single cells. The authors characterize the underlying genetics, spike binding, neutralization and epitopic distributions of thousands of memory B cells from human donors following vaccination. They find that infection followed by vaccination elicits a more robust memory B cell response, that is more able to bind and neutralize variants of concern than immunization alone. The hierarchy of epitopes that are engaged and the genetics of those B cells differ between cohorts. These results are an important step towards linking ambiguous qualities of immune serum to precise phenotypes and genetics of individual B cells/antibodies.

My concern is that the timing of sample collection differs between cohorts and may capture two different phases of the immune response. The seronegative samples were collected an average of ~50 days following the second vaccine dose while the seropositive samples were collected an average ~21 days after the second vaccine dose (one person donated a sample before a second shot). The seropositive cohort would be at or near the peak of MBC circulation in peripheral blood while the seronegative cohort would be well past this peak. This discrepancy would also apply to serum antibody titers. Thus, direct comparisons between the two cohorts are confounded by this difference, in particular in the number and viability of the cell populations. I note that the conclusions themselves are intuitive and consistent with other reports. The text should acknowledge the differences between cohorts through the text.

For the reasons outlined above I am not sure that statistics should be applied between the two cohorts. Because comparisons and conclusions are descriptive, and at times based on visually small differences, additional attention to how the results are presented to the reader would be beneficial.

Minor:

It would be more accurate to refer to the MBC population that was characterized as "circulating" or "peripheral" throughout the text.

Extended Data Table 1. Columns indicating the period of time between infection and first immunization as well as time between the second immunization and sample collection should be added.

Gene utilization plots would be clearer if they were presented with bars rather than points connected with a trend line. Since these are individual data points a trend line may not be appropriate.

Line 53: Use of "Molecular dynamics". This term has other biological/chemical connotations please consider a different term.

Referee #2 (Remarks to the Author):

In this manuscript by Andreano et al., the authors analyzed memory B cell responses in convalescent (n=5) and naïve (n=5) individuals who received SARS-CoV-2 Pfizer's mRNA vaccine. Blood specimens were collected around one month after last immunization. Spike+ memory B cells were single cell sorted and corresponding monoclonal antibodies (mAbs) were tested by ELISA and neutralization against the originally circulating strain and emerging variants. Impressively, the authors have assessed 3,200+ spike-specific mAbs from the 10 subjects. Main conclusion is that the memory B cell response in the convalescent individuals was quantitatively and qualitatively superior. Overall, this study provides a very comprehensive characterization of the anti-spike memory B cell response after SARS-CoV-2 mRNA vaccination in humans. The text and figures are clear.

- L189 (Discussion): "This conclusion is not surprising since several papers have already reported that vaccination of convalescent people induces a hybrid immunity with titers of neutralizing antibodies up to 50-fold higher than those induced by vaccination of naïve people."

This is exactly my major observation. As mentioned above, the work is exceptionally detailed and the number of mAbs analyzed is astounding. However, the work is basically comparing the memory B cell response in individuals who were exposed to the spike protein three times vs. those who were exposed twice. It would have been much more informative if such comprehensive comparison was made between memory B cells isolated from SARS-CoV-2 infected vs. SARS-CoV-2 vaccinated individuals.

- Spike+ memory B cell clones from the convalescent individuals accumulated greater number of somatic hypermutations (Ext. Fig. 4b). This has likely enhanced the efficiency of capturing these cells with the fluorescent spike probes (quantitative impact) and the efficiency of the mAbs to neutralize the virus (qualitative impact).

- The Authors did not examine the clonality of the generated mAbs. It could be that many of the examined mAbs are members of the same clonal pool.

- Extended Fig. 1 data lacking a negative control showing the background levels of spike staining of memory B cells from blood samples collected before the pandemic.

- Minor: The critical information about when the blood was collected after last immunization cannot be found in the text, figure, figure legend or even the methods section describing "human sample collection". One had to look at extended data table 1 to find this piece of information.

Author Rebuttals to Initial Comments:

Referee #1:

Andreano and colleagues compare, at the single B cell level, the antibody response to SARS-CoV-2 in donors that were infected and then vaccinated to donors that were seronegative at the time of vaccination. While others have very recently reported that infection followed by vaccination can elicit antibodies that bind to spike proteins more avidly and more broadly than vaccination alone, the authors have extended these observations to the level of single cells. The authors characterize the underlying genetics, spike binding, neutralization and epitopic distributions of thousands of memory B cells from human donors following vaccination. They find that infection followed by vaccination elicits a more robust memory B cell response, that is more able to bind and neutralize variants of concern than immunization

alone. The hierarchy of epitopes that are engaged and the genetics of those B cells differ between cohorts. These results are an important step towards linking ambiguous qualities of immune serum to precise phenotypes and genetics of individual B cells/antibodies.

R1: My concern is that the timing of sample collection differs between cohorts and may capture two different phases of the immune response. The seronegative samples were collected an average of ~50 days following the second vaccine dose while the seropositive samples were collected an average ~21 days after the second vaccine dose (one person donated a sample before a second shot). The seropositive cohort would be at or near the peak of MBC circulation in peripheral blood while the seronegative cohort would be well past this peak. This discrepancy would also apply to serum antibody titers. Thus, direct comparisons between the two cohorts are confounded by this difference, in particular in the number and viability of the cell populations. I note that the conclusions themselves are intuitive and consistent with other reports. The text should acknowledge the differences between cohorts through the text. For the reasons outlined above I am not sure that statistics should be applied between the two cohorts. Because comparisons and conclusions are descriptive, and at times based on visually small differences, additional attention to how the results are presented to the reader would be beneficial.

A1: We thank the referee for the comment. The authors were aware that the timing between the two cohorts was not exactly the same. However, we feel confident about our data because the serology data are in line with several published papers reporting the serum activity (binding and neutralization) of similar populations^{1,2}. In addition, it is still possible to observe a clear difference among seronegative and seropositive subjects when memory B cells IgM⁻ S protein⁺ subsets were compared at similar timing for blood collection. Examples are subjects VAC-002 (seronegative) and VAC-006 (seropositive) where blood was collected for both of them at around 30 days. Another example are subjects VAC-010 (seronegative) and VAC-003 or VAC-005 (seropositives) where blood was collected at around 20 days. An overall average around 3-fold higher IgM⁻ S protein⁺ B cells was observed in the seropositive donors. Anyway, following the suggestions of the reviewer, we acknowledged the differences between cohorts through the text at page 5, line 61-63 and modified the extended data table 1 to make this difference between analyzed groups more clear.

Minor:

R2: It would be more accurate to refer to the MBC population that was characterized as “circulating” or “peripheral” throughout the text.

A2: Circulating or peripheral was added throughout the text at page 5, line 61 – 64.

R3: Extended Data Table 1. Columns indicating the period of time between infection and first immunization as well as time between the second immunization and sample collection should be added.

A3: We modified the Extended Data Table 1 as suggested by the reviewer.

R4: Gene utilization plots would be clearer if they were presented with bars rather than points connected with a trend line. Since these are individual data points a trend line may not be appropriate.

A4: We tried to modify the figure in accordance with the reviewer suggestion but in our mind it was more difficult to visualize and interpret the data. Therefore, we kept the original figure.

R5: Line 53: Use of “Molecular dynamics”. This term has other biological/chemical connotations please consider a different term.

A5: In accordance with the reviewer suggestion, “molecular dynamics” was substituted with “molecular mechanisms”.

Referee #2:

In this manuscript by Andreano et al., the authors analyzed memory B cell responses in convalescent (n=5) and naïve (n=5) individuals who received SARS-CoV-2 Pfizer’s mRNA vaccine. Blood specimens was collected around one month after last immunization. Spike+ memory B cells were single cell sorted and corresponding monoclonal antibodies (mAbs) were tested by ELISA and neutralization against the originally circulating strain and emerging variants. Impressively, the authors have assessed 3,200+ spike-specific mAbs from the 10 subjects. Main conclusion is that the memory B cell response in the convalescent individuals was quantitatively and qualitatively superior. Overall, this study provides a very comprehensive characterization of the anti-spike memory B cell response after SARS-CoV-2 mRNA vaccination in humans. The text and figures are clear.

R1: L189 (Discussion): "This conclusion is not surprising since several papers have already reported that vaccination of convalescent people induces a hybrid immunity with titers of neutralizing antibodies up to 50-fold higher than those induced by vaccination of naïve people." This is exactly my major observation. As mentioned above, the work is exceptionally detailed and the number of mAbs analyzed is astounding. However, the work is basically comparing the memory B cell response in individuals who were exposed to the spike protein three times vs. those who were exposed twice. It would have been much more informative if such comprehensive comparison was made between memory B cells isolated from SARS-CoV-2 infected vs. SARS-CoV-2 vaccinated individuals.

A1: We thank the reviewer for rising this point. We did not include infected people in this study because B cells from infected people have been already studied by many laboratories worldwide³⁻⁶. The intent of our study was to answer pragmatic questions raised by vaccination and whether vaccination after infection induces a different quality of immune response which may inform policy makers. In fact, we believe that our data can be of extreme value especially in this specific moment where a decision about third vaccination doses has to be taken.

R2: Spike+ memory B cell clones from the convalescent individuals accumulated greater number of somatic hypermutations (Ext. Fig. 4b). This has likely enhanced the efficiency of capturing these cells with the fluorescent spike probes (quantitative impact) and the efficiency of the mAbs to neutralize the virus (qualitative impact).

A2: We included this point in our discussion (page 10, line 183 – 186) to highlight that a higher qualitative and quantitative impact in seropositive people may derive from the overall higher somatic hypermutation levels observed in this cohort.

R3: The Authors did not examine the clonality of the generated mAbs. It could be that many of the examined mAbs are members of the same clonal pool.

A3: We previously examined the clonality of generated mAbs and we did not report any data as no major clonal families were found to be expanded in our repertoire. Anyway, to highlight this point into the text, we included a sentence in our manuscript at page 7, line 131-132.

R4: Extended Fig. 1 data lacking a negative control showing the background levels of spike staining of memory B cells from blood samples collected before the pandemic.

A4: We modified Extended Data Figure 1 in order to include a negative control as suggested by the reviewer.

R5: Minor: The critical information about when the blood was collected after last immunization cannot be found in the text, figure, figure legend or even the methods section describing “human sample collection”. One had to look at extended data table 1 to find this piece of information.

A5: We added a sentence to include the data in the text (page 5, line 61-62) and adjusted Extended Data Table 1 to improve clarity as suggested by the reviewer.

REFERENCES

1. Krammer, F., *et al.* Antibody Responses in Seropositive Persons after a Single Dose of SARS-CoV-2 mRNA Vaccine. **384**, 1372-1374 (2021).
2. Wang, Z., *et al.* Naturally enhanced neutralizing breadth against SARS-CoV-2 one year after infection. *Nature* **595**, 426-431 (2021).
3. Gaebler, C., *et al.* Evolution of antibody immunity to SARS-CoV-2. *Nature* **591**, 639-644 (2021).
4. Robbiani, D.F., *et al.* Convergent antibody responses to SARS-CoV-2 in convalescent individuals. *Nature* **584**, 437-442 (2020).
5. Sokal, A., *et al.* Maturation and persistence of the anti-SARS-CoV-2 memory B cell response. *Cell* **184**, 1201-1213.e1214 (2021).
6. Sakharkar, M., *et al.* Prolonged evolution of the human B cell response to SARS-CoV-2 infection. **6**, eabg6916 (2021).

Reviewer Reports on the First Revision:

Referee #2 (Remarks to the Author):

My main concern that the work does not conceptually advance the field in a significant manner remains unaddressed.